# $k$NN Attention Demystified: A Theoretical Exploration for Scalable Transformers

**Themistoklis Haris**
Boston University
tharis@bu.edu

## Abstract

Despite their power, Transformers (Vaswani, 2017) face challenges with long sequences due to the quadratic complexity of self-attention. To address this limitation, methods like $k$-Nearest-Neighbor ($k$NN) attention have been introduced (Roy et al., 2021), enabling each token to attend to only its $k$ closest tokens. While $k$NN attention has shown empirical success in making Transformers more efficient, its exact approximation guarantees have not been theoretically analyzed. In this work, we establish a theoretical framework for $k$NN attention, reformulating self-attention as expectations over softmax distributions and leveraging lazy Gumbel sampling (Mussmann et al., 2017) with $k$NN indices for efficient approximation. Building on this framework, we also propose novel sub-quadratic algorithms that approximate self-attention gradients by leveraging efficient sampling techniques, such as Markov Chain-based estimation. Finally, we demonstrate the practical effectiveness of these algorithms through empirical experiments, showcasing their benefits in both training and inference.

## 1 Introduction

Transformer models have become the dominant neural architecture across language, vision, and other domains (Vaswani, 2017; Dosovitskiy et al., 2020). However, scaling them to handle larger input sequences remains a significant challenge (Tay et al., 2020), primarily due to the quadratic complexity of computing self-attention. Overcoming this limitation is crucial for advancing neural networks. Extending context length would enable Transformers to tackle complex tasks like book summarization (Kryściński et al., 2021) and time-series forecasting (Wen et al., 2022; Zeng et al., 2023; Zhou et al., 2021). Furthermore, improving attention efficiency would reduce the computational burden of training, making these models more accessible. Bridging this "compute divide" is vital for democratizing AI (Ahmed & Wahed, 2020).

Efficient computation of self-attention has been a focal point of research in recent years (Fournier et al., 2023). Flash Attention (Dao et al., 2022) and related work (Saha & Ye, 2024) optimize the exact calculation of attention by minimizing wasted computation during GPU I/O operations. However, most approaches focus on approximating the attention function. Sparse Transformers improve efficiency by allowing each token to attend to only a small subset of tokens (Meister et al., 2021). These subsets are identified through deterministic methods (Child et al., 2019; Guo et al., 2019; Soldaini & Moschitti, 2020; Li et al., 2019; Qiu et al., 2019; Beltagy et al., 2020; Chen et al., 2021), randomized algorithms (Kitaev et al., 2020; Han et al., 2023; Zandieh et al., 2023; Pagliardini et al., 2024), or adaptive techniques (Correia et al., 2019). Additionally, self-attention is often approximated using low-rank matrices and kernel methods (Wang et al., 2020; Tay et al., 2021; Xiong et al., 2021; Katharopoulos et al., 2020; Choromanski et al., 2020). On the negative side, recent fine-grained complexity reductions indicate that achieving a good approximation with sub-quadratic time is not feasible across all scenarios (Keles et al., 2023; Alman & Song, 2024a).

In this work, we focus on sparse attention methods where each token vector $q_i \in \mathbb{R}^d$ attends to the $k$ tokens $k_j \in \mathbb{R}^d$ with the largest inner products $q_i^T k_j$ (Gupta et al., 2021; Wang et al., 2022), a paradigm we refer to as *$k$NN Attention*. The Routing Transformer (Roy et al., 2021) was an early example, using $k$-means clustering to ensure each query only attends to keys within the same cluster. Memorizing Transformers (Wu et al., 2022) later extended this approach by leveraging $k$NN search

within a stored memory, enabling models to memorize new data during inference. More recently, Unlimiformer models (Bertsch et al., 2024) have improved efficiency by using a single $k$NN data structure (or *index*) across all attention heads and layers.

Previous works have empirically shown that $k$NN Attention not only improves computational efficiency, but also enhances model architectures and capabilities. However, a rigorous theoretical analysis of $k$NN Attention is still lacking. Key questions remain unresolved, including the precise approximation guarantees it offers, the optimal value of $k$, and whether these methods can also help to yield efficient algorithms for approximating the backward pass.

For a comprehensive outline of preliminary results and theory, please refer to Appendix A.

**Notation**  Let $Q, K, V \in \mathbb{R}^{n \times d}$ be our *query, key* and *value* matrices. Let $q_i = Q_{i,:} \in \mathbb{R}^d$ be $i$-th *row* of $Q$ written as a column vector. We will also denote the $j$-th column of $Q$ by $Q_{:,j}$. We define $A := QK^T \in \mathbb{R}^{n \times n}$ to be the *attention matrix*, and $O = \text{softmax}(A) \cdot V \in \mathbb{R}^{n \times d}$ to be the output of the *attention function*. The softmax function is applied row-wise to $A$ and is defined as a vector valued function $\sigma : \mathbb{R}^n \to \mathbb{R}^n$:

$$\sigma(y_1, ..., y_n)_i = \frac{\exp(y_i)}{\sum_{s=1}^{n} \exp(y_s)} \tag{1}$$

We also let $[n] := \{1, 2, ..., n\}$ and use the notation $\text{polylog}(n)$ as a substitute of $\log^k(n)$ for some arbitrary constant $k \in \mathbb{Z}^+$ that is independent of $n$. Finally, we use the $\widetilde{O}$ notation to hide polylogarithmic factors. We design randomized, Monte Carlo algorithms that can fail with probability $\delta > 0$. We will often make use of the following boosting lemma:

**Lemma 1.1** (Median-Of-Means Boosting, Chakrabarti (2020)). *If $\widehat{Q}$ is an unbiased estimator of some statistic, then one can obtain an $(\varepsilon, \delta)$-multiplicative estimate of that statistic by suitably combining $K := \frac{C}{\varepsilon^2} \frac{Var[\widehat{Q}]}{\mathbb{E}[\widehat{Q}]^2} \ln \frac{2}{\delta}$ independent samples of $\widehat{Q}$, where $C$ is a universal constant.*

### 1.1  OUR CONTRIBUTIONS AND RESULTS

**A Theoretical Framework for $k$NN Attention**  Our work provides a theoretical framework to explain both the efficiency and effectiveness of $k$NN Attention. Our framework reformulates self-attention as expectations over softmax distributions. These expectations are approximated by sampling from each distribution in sublinear time using Lazy Gumbel Noise Sampling. By connecting $k$NN, $k$-Maximum Inner Product Search (MIPS), and Gumbel noise sampling, we develop a new sub-quadratic self-attention approximation algorithm aligning with the $k$NN Attention paradigm with precise theoretical guarantees, given in full as Theorem 2.4.

**Approximating the Backward Pass**  Our framework can be extended to solve the problem of approximating attention gradients. Even though backpropagation is the main memory bottleneck for large models, few methods approximate attention gradients directly. The work of Alman & Song (2024a) is most relevant, deriving inapproximability results for certain parameter regimes.

We present new approximation algorithms for self-attention gradients using $k$NN search. Our results are encapsulated in full by Theorem 3.1. A key challenge is the need to multiply by the transpose of a stochastic matrix, which disrupts our expectation-based reformulation. To address this, we use a Markov-Chain sampling technique, treating the attention matrix as a transition matrix and applying a single-step iteration. We also employ other sampling techniques, such as CDF-based sampling, in novel ways to achieve sub-quadratic time complexity.

## 2  $k$NN ATTENTION AS AN APPROXIMATION ALGORITHM

We begin by viewing the self-attention output as a matrix of expectations under various softmax distributions. This reformulation has been used in prior works (Kratsios, 2021; Singh & Buckley, 2023) but, to our knowledge, has not been used for analyzing sparse attention mechanisms. Let $D_i$ be the softmax distribution defined by $D_i(j) \propto \exp(q_i^T \cdot k_j)$ over $[n]$. Then, notice that we can

write:

$$O_{ij} = \sum_{r=1}^{n} \frac{\exp(q_i^T k_r)}{\sum_{s=1}^{n} \exp(q_i^T k_s)} \cdot V_{rj} = \sum_{r=1}^{n} D_i(r) \cdot V_{rj} = \underset{r \sim D_i}{\mathbb{E}} [V_{rj}] \qquad (2)$$

Thus, to approximate $O_{ij}$, we have to estimate the expected value in Equation 2. Let $r$ be sampled according to $D_i$. Then, the estimator $\widehat{O}_{ij} = V_{rj}$ is unbiased, as $\mathbb{E}_{r \sim D_i}[\widehat{O}_{ij}] = O_{ij}$. Assuming an upper bound on $||V||_\infty$, and that the values $O_{ij}$ do not get arbitrarily small[1], we can bound the variance of $\widehat{O}_{ij}$ and use boosting to obtain explicit multiplicative error guarantees:

**Theorem 2.1.** *Suppose $||V||_\infty \leq B = O(\log(n))$ and assume that for any $i \in [n]$ we can sample from $D_i$ in expected time $O(T)$. Further, we assume that for some constant $C$, it holds that $O_{ij} \geq C$ for all $i, j$. Then, there exists an algorithm to output a matrix $\widehat{O} \in \mathbb{R}^{n \times d}$ such that:*

$$|\widehat{O}_{ij} - O_{ij}| \leq \varepsilon O_{ij} \qquad (3)$$

*for all $(i, j) \in [n] \times [d]$ with probability at least $1 - \delta$, where $\varepsilon, \delta > 0$ are constants. The algorithm runs in $O(nd \cdot T \cdot \varepsilon^{-2} \log(nd/\delta) \log n)$ time in expectation.*

*Proof.* Given that $\widehat{O}_{ij}$ is an unbiased estimator of $O_{ij}$, we can utilize Lemma 1.1 to get an $(\varepsilon, \delta)$-multiplicative estimator for $O_{ij}$. To determine a sufficient number of samples of $\widehat{O}_{ij}$, we first bound the variance of our estimator:

$$\text{Var}\left[\widehat{O}_{ij}\right] \leq \underset{r \sim D_i}{\mathbb{E}} \left[V_{rj}^2\right] = \sum_{r=1}^{n} D_i(r) V_{rj}^2 \leq B \cdot O_{ij} \qquad (4)$$

Then, the number of samples required is:

$$O\left(\varepsilon^{-2} \cdot \log(1/\delta) \cdot \text{Var}\left[\widehat{O}_{ij}\right] \cdot \mathbb{E}\left[\widehat{O}_{ij}\right]^{-2}\right) = O\left(\varepsilon^{-2} \cdot \log(1/\delta) \log n\right) \qquad (5)$$

due to our assumption on $||V||_\infty$ and the lower bound on $O_{ij}$. To ensure that all $nd$ elements of $O$ are approximated within the desired guarantees, we have to set $\delta' := \delta/(nd)$ and union-bound over all $nd$ elements of $O$. Since each sample requires $O(T)$ time, we arrive at the desired time complexity. $\qquad \square$

## 2.1 Efficient Sampling from $D_i$ via Lazy Gumbel Sampling

Theorem 2.1 previously assumed we could directly sample from the distribution $D_i$ in time $O(T)$. The *Lazy Gumbel Sampling* method proposed by Mussmann et al. (2017) provides a way to sample from each $D_i$ in sublinear time, even with limited knowledge of $D_i$. However, there is an initial pre-processing step that takes a bit more than linear time across all the distributions. In this section we present this method and clarify a value for $T$ to be used in Theorem 2.1.

Fix some $i \in [n]$ and let $Z_{ij} = q_i^T k_j$. In the Gumbel Max Trick (Lemma A.3), we form the random variables $N_{ij} = Z_{ij} + G_{ij}$ where $G_{ij} \sim \text{Gumbel}(0, 1)$ for all $j \in [n]$ and sample $\arg\max N_{ij}$. This is equivalent to sampling $j \in [n]$ from the softmax distribution over the $Z_{ij}$ scores. Mussmann et al. (2017) observed that if we have the top $k$ $Z_{ij}$ values in a set $S_i$ and add Gumbel noise to just them, then for any $j \notin S_i$ to be ultimately picked, its Gumbel noise $G_{ij}$ must be quite large. We can use the concentration properties of the Gumbel distribution to argue that in expectation we only need to sample $\frac{n}{k}$ elements not in $S_i$. Setting $k = \sqrt{n}$ allows us to balance the two, resulting in a sublinear time algorithm for sampling from $D_i$. An illustration of the idea can be seen in Figure 1, as it was presented in Mussmann et al. (2017). We can see that this method samples exactly from $D_i$.

**Theorem 2.2** (Correctness of Algorithm 1, (Mussmann et al., 2017))**.** *After running Algorithm 1, it holds that:*

$$\widehat{j} = \arg\max_{j \in [n]} \{q_i^T k_j + G_{ij}\} \qquad (6)$$

*where $G_{ij} \sim \text{Gumbel}(0, 1)$. In other words, $\widehat{j}$ is sampled according to $D_i$.*

---

[1]If $O_{ij}$ is arbitrarily close to 0, then we can achieve low additive error by just outputting $\widehat{O}_{ij} = 0$.

---

**Algorithm 1** Lazy Gumbel Sampling from $D_i$, for some $i \in [n]$, (Mussmann et al., 2017)

---

1: **Inputs:** $k \in \mathbb{N}$, $q_i \in \mathbb{R}^d$, $K \in \mathbb{R}^{n \times d}$, $S_i := \{$the $k$ keys $j$ with the largest $Z_{ij} := q_i^T k_j\}$.
2: Sample $G_{ij} \sim \text{Gumbel}(0, 1)$ for $j \in S_i$.
3: Let $M \leftarrow \max_{j \in S_i}\{Z_{ij} + G_{ij}\}$ and $S_{\min} \leftarrow \min_{j \in S_i}\{Z_{ij}\}$.
4: Let $B \leftarrow M - S_{\min}$ be the Gumbel cutoff.
5: Let $m \sim \text{Bin}(n - k, 1 - \exp(-\exp(-B)))$ be the number of $[n] \setminus S_i$ Gumbels greater than $B$. Sample $m$ points from $[n] \setminus S_i$ and denote the set of sampled points as $T_i$.
6: Sample $G_{ij} \sim \text{Gumbel}(0, 1)$ conditionally greater than $B$ for each $j \in T_i$.
7: **return** $\widehat{j} \leftarrow \arg \max_{j \in S_i \cup T_i}\{Z_{ij} + G_{ij}\}$

---

*Proof.* The only way that we do not find the maximum is if one of the points in $[n] \setminus (S_i \cup T_i)$ are the true maximum. However those points (by construction) have Gumbel noise at most $B$, so they cannot be the overall maximum. $\square$

In Appendix B, we show that the expected number $m$ of large Gumbels is at most $n/k$. Our simplified proof uses the Gumbel distribution's Moment Generating Function, rather than the original exponential-based analysis.

**Lemma 2.1** ((Mussmann et al., 2017))*. The following holds:*

$$\mathbb{E}[m] \leq \frac{n}{k} \qquad (7)$$

Due to Lemma 2.1, we see that we need to set $k = \sqrt{n}$ to optimize our overall time complexity. As a result, by combining the pseudocode of Algorithm 1 and Lemma 2.1, Mussmann et al. (2017) arrive at the following:

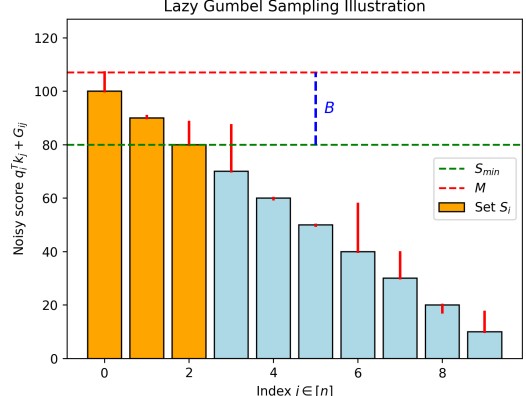

Figure 1: Lazy Gumbel sampling

**Theorem 2.3.** *Let $k = \sqrt{n}$. Suppose that we are able to retrieve the set $S_i$ in $f(n, k)$ time. Then, we can use Algorithm 1 to sample from $D_i$ in $O(\sqrt{n} + f(n, \sqrt{n}))$ time in expectation.*

### 2.1.1 OBTAINING THE TOP $k$ INNER PRODUCTS

Algorithm 1 relies on obtaining the set $S_i$ of the top $\sqrt{n}$ inner products $q_i^T k_j$ for each $i \in [n]$ in sub-quadratic $f(n, \sqrt{n})$ time. Since the $k_j$ vectors are fixed, while the $q_i$ vectors act as queries, this setup is known as the $k$**-Maximum Inner Product Search Problem (MIPS)**

The $k$-MIPS problem can be reduced to the $k$NN problem using a transformation proposed by Neyshabur & Srebro (2015). We add an extra dimension to normalize all key vectors. Specifically, the inner product $q_i^T k_j$ can be expressed as:

$$q_i^T k_j = \frac{1}{2}\left(||q_i||_2^2 + ||k_j||_2^2 - ||q_i - k_j||_2^2\right) \qquad (8)$$

If the norms $||k_j||_2$ are the same across all $j$, the problem reduces to finding the $k$ nearest neighbors to $q_i$. To enforce this, we define:

$$(k_j')^T = \left[k_j^T, \sqrt{M - ||k_j||_2^2}\right] \qquad (9)$$

so that $||(k')_j||_2 = M$ for all $j \in [n]$, where $M$ is a previously known upper bound. When querying with $q_i$, we use:

$$(q_i')^T = \left[q_i^T, 0\right] \qquad (10)$$

This transformation preserves the inner products, allowing us to solve the $k$NN problem for $q_i'$. We can then use a $k$NN index $H$ to preprocess $K$ and query it with each $q_i'$ to construct $S_i$ for all $i \in [n]$. We remain agnostic to the specific $k$NN index one could use for this algorithm[2], but if we assume that the construction runtime is slightly larger than linear and the query time slightly larger than $k$, then Theorem 2.1 with $T \approx O(\sqrt{n})$ give us a total runtime of $\approx \widetilde{O}(dn^{3/2} \cdot \varepsilon^{-2} \log(1/\delta))$.

---

**Algorithm 2** $k$NN Attention

---

1: **Inputs:** $Q, K, V \in \mathbb{R}^{n \times d}$, error parameter $\varepsilon > 0$, confidence parameter $\delta > 0$, $k \in \mathbb{N}$.
2: **for** $j \in [n]$ **do**                                                   ▷ Pre-Processing: Lines 2-4
3:     $(k_j')^T = \left[ k_j^T, \sqrt{M - ||k_j||_2^2} \right] \in \mathbb{R}^{(d+1) \times 1}$
4: $H \leftarrow$ Build a $k$NN index from $\{k_j' \mid j \in [n]\}$
5: **for** $i \in [n]$ **do**
6:     $(q_i')^T \leftarrow [q_i^T, 0] \in \mathbb{R}^{d+1}$
7:     Query $H$ with $q_i'$ to get $S_i$ with $|S_i| = k$.
8:     **for** $j \in [d]$ **do**
9:         $\widehat{O}_{ij} \leftarrow$ Median-Of-Means with Algorithm 1 as sampler $\leftarrow (k, q_i, K, S_i)$.
10: **return** $\widehat{O}$

---

Overall, our framework for $k$NN Attention can be summarized by the following theorem:

**Theorem 2.4.** *Let* $Q, K, V \in \mathbb{R}^{n \times d}$ *and* $\varepsilon, \delta \in (0,1)$ *be positive constants. Assume* $||V||_\infty = O(\log n)$. *Then, Algorithm 2 with* $k = \sqrt{n}$ *outputs a matrix* $\widehat{O} \in \mathbb{R}^{n \times d}$ *such that:*

$$|\widehat{O}_{ij} - O_{ij}| \leq \varepsilon O_{ij} \tag{11}$$

*for all* $(i, j) \in [n] \times [d]$ *with probability at least* $1 - \delta$ *and in expected sub-quadratic time and space.*

*Proof.* Algorithm 2 approximates each entry of the attention matrix by using Median-of-Means sampling. Since $k = \sqrt{n}$, the sampling from each softmax distribution is done using Algorithm 1 according to the guarantees of Theorem 2.3. Theorem 2.1 thus gives us the desired runtime. ☐

## 2.2 $k$NN ATTENTION WITHOUT MEDIAN-OF-MEANS

This section describes a simpler algorithm for computing the expected value needed for self-attention. The algorithm still uses $k$NN indices to find the top $k$ inner products per query, but usually outperforms Algorithm 2 in practice, due to its amenity for hardware-accelerated vectorization, and is thus our preferred implementation for experiments[3].

Building on Mussmann et al. (2017), we estimate $\mathbb{E}_{k \sim D_i}[V_{kj}]$ using set $S_i$ by sampling $\ell$ additional vectors outside $S_i$ (set $T_i$) and upweighting them in the expectation sum, as follows:

$$\widehat{O}_{ij} = \frac{\sum_{s \in S_i} e^{q_i^T k_s} \cdot V_{sj} + \frac{n-k}{\ell} \sum_{s \in T_i} e^{q_i^T k_s} \cdot V_{sj}}{\sum_{s \in S_i} e^{q_i^T k_s} + \frac{n-k}{\ell} \sum_{s \in T_i} e^{q_i^T k_s}} \tag{12}$$

The quality of this estimator and the optimal choices for $k$ and $\ell$ are derived as follows:

**Theorem 2.5.** *Assuming* $||V||_\infty \leq C$, *the estimator* $\widehat{O}_{ij}$ *satisfies the following error guarantee with probability at least* $1 - \delta$:

$$\left| \widehat{O}_{ij} - O_{ij} \right| \leq \varepsilon C \tag{13}$$

*if the following two conditions hold:* $k^2 \ell \geq 8n^2 \varepsilon^{-2} \log(4/\delta)$ *and* $k\ell \geq 2n\varepsilon^{-2} \log(2/\delta)$. *Setting* $k = \ell = O\left(n^{2/3} \varepsilon^{-1} \sqrt{\log(1/\delta)}\right)$ *gives us an* $\widetilde{O}\left(dn^{5/3} \varepsilon^{-1} \sqrt{\log(1/\delta)}\right)$ *algorithm for estimating self-attention within additive error* $O(\varepsilon)$, *assuming an efficient* $k$NN *implementation.*

---

[2]For a specific construction with precise theoretical guarantees that uses Locality Sensitive Hashing (LSH), please refer to Appendix C.

[3]See Appendix G for a PyTorch implementation of this algorithm.

*Proof.* The proof of the additive error guarantee can be found in Mussmann et al. (2017). ☐

## 3 ATTENTION GRADIENT ESTIMATION VIA MARKOV CHAIN SIMULATIONS

Next, we present randomized algorithms which can efficiently approximate the gradients of the self-attention function. First, we give exact formulas for the gradients in question. These can be obtained by applying the chain rule repeatedly, as shown in the proof of Lemma 3.1, in Appendix D.

**Lemma 3.1** (Attention Gradients). *Let $Q, K, V \in \mathbb{R}^{n \times d}$. Let $P := softmax\left(QK^T\right) \in \mathbb{R}^{n \times n}$ be the normalized attention matrix. Let $\phi$ be a differentiable scalar function of $O$ and $D^O = \partial\phi/\partial O \in \mathbb{R}^{n \times d}$. Similarly define $D^Q, D^K$ and $D^V$. The following relationships hold:*

$$D^V = P^T \cdot D^O \tag{14}$$

$$D_{ij}^Q = \sum_{k=1}^{n} P_{ik} \left(D_{ik}^P - \langle D_{i,:}^P, P_{i,:} \rangle\right) K_{kj} \tag{15}$$

$$D_{ij}^K = \sum_{k=1}^{n} P_{ki} \left(D_{ki}^P - \langle D_{k,:}^P, P_{k,:} \rangle\right) Q_{kj} \tag{16}$$

*where $D_{ij}^P := \partial\phi/\partial P_{ij} = \langle D_{i,:}^O, V_{j,:} \rangle$.*

Clearly, calculating $D^Q, D^K$ and $D^V$ can be done in $O(dn^2)$ time. We propose algorithms for estimating these gradients that run in sub-quadratic time. We first present Algorithms 3 and 4 for the estimation of $D^V$, whereas our algorithms for estimating $D^K$ and $D^Q$ can be found in Appendices F and E. Ultimately, we prove the following theorem:

**Theorem 3.1.** *Let $\phi$ be a differentiable scalar loss function and $\partial\phi/\partial O \in \mathbb{R}^{n \times d}$. Then, under certain assumptions on the $||\cdot||_\infty$ norms of $Q, K, V, D^O$, there exist sub-quadratic time algorithms that output estimates $\widehat{D}^Q, \widehat{D}^K, \widehat{D}^V \in \mathbb{R}^{n \times d}$ for which with probability at least $1 - \delta$ it holds that:*

$$||\widehat{D}^Q - \partial\phi/\partial Q||_\infty \leq e_Q, \ ||\widehat{D}^K - \partial\phi/\partial K||_\infty \leq e_K \ and \ ||\widehat{D}^V - \partial\phi/\partial V||_\infty \leq e_V \tag{17}$$

*where $e_Q, e_K, e_V$ are explicit error parameters that can roughly be bounded by $O(\varepsilon n \cdot polylog(n))$.*

*Proof.* We combine the guarantees of Theorems 3.2, F.1 and E.1, where the assumptions on the $||\cdot||_\infty$ norms of $Q, K, V$ and $D^O$ can also be found. ☐

### 3.1 ESTIMATING $D^V$

**Theorem 3.2.** *Given $Q, K, V$ and $D^O$, Algorithm 4 calculates $\partial\phi/\partial V_{ij} = D_{ij}^V$ for all $(i, j) \in [n] \times [d]$ within an additive approximation error of*

$$e_V = \varepsilon \cdot \langle D_{:,j}^O, 1^n \rangle + 2n\varepsilon M_j, \ where \ M_j := - \min_{i \in [n], D_{ij}^O \leq 0} D_{ij}^O \tag{18}$$

*with probability at least $1 - \frac{1}{n}$. The time complexity is $O(nd^2\varepsilon^{-2}\log n)$.*

*Proof.* Suppose we want to calculate the $j$-th column of $D^V$:

$$D_{:,j}^V = P^T \cdot D_{:,j}^O \tag{19}$$

for $j \in [d]$. Fix $\overrightarrow{x_j} := D_{:,j}^O \in \mathbb{R}^{n \times 1}$ and suppose that $\overrightarrow{x_j} \geq 0$. We will soon explicitly relax this assumption. Then, $\overrightarrow{y_j} := \overrightarrow{x_j}/||\overrightarrow{x_j}||_1$ is a distribution over the universe $[n]$. Imagine a random walk over $[n]$ with transition matrix $P$ and initial distribution $\overrightarrow{y_j}$. Then:

$$\overrightarrow{\pi_j} := P^T \cdot \overrightarrow{y_j} \tag{20}$$

is the distribution after one step in the process. Thus, we can estimate $\overrightarrow{\pi_j}$ with Markov Chain simulations, by first picking an item $i \in [n]$ from the distribution $\overrightarrow{y_j}$, and then picking another item $k \in [n]$ with probability $P_{ik}$. We make $N$ independent length-1 random walks like this and let:

$$X_v^{(j,s)} = \begin{cases} 1, & \text{if the } s\text{-th walk ends up in state } v \\ 0, & \text{otherwise} \end{cases} \tag{21}$$

We know that $\mathbb{E}[X_v^{(j,s)}] = \pi_j(v)$ for all $s \in [N]$. Thus, we can form a boosted estimator $\widehat{p}_j(v) = \frac{1}{N} \sum_{s=1}^N X_v^{(j,s)}$. This estimator is unbiased due to linearity of expectation, so we can use the Hoeffding bound to ensure that our empirical distribution is close to the true distribution as long as we take enough samples:

$$\Pr\left[|\widehat{p}_j(v) - \pi_j(v)| \geq \varepsilon\right] \leq 2\exp(-2N\varepsilon^2)$$

We set the probability of failure to $1/(dn^2)$ so that we can union bound over all $v \in [n]$ and all $j \in [d]$. It follows that we require:

$$N = \Theta(\varepsilon^{-2}\ln(nd)) \tag{22}$$

Of course, we need to scale $\overrightarrow{\pi_j}$ back to recover $D_{:,j}^V$. We define:

$$\widehat{D}_{:,j}^V = ||\overrightarrow{x_j}||_1 \cdot \widehat{p}_j \tag{23}$$

Then we get that with probability at least $1 - 1/n$ it holds for all $j \in [d]$ that:

$$\left|\left|\widehat{D}_{:,j}^V - D_{:,j}^V\right|\right|_\infty = ||\overrightarrow{x_j}||_1 \cdot ||\widehat{p}_j - \overrightarrow{\pi_j}||_\infty \leq \varepsilon||\overrightarrow{x_j}||_1 \tag{24}$$

**Relaxing the non-negativity assumption** We now relax the non-negativity constraint on $\overrightarrow{x_j}$, accepting some approximation error. Since normalizing $\overrightarrow{x_j}$ with its L1 norm fails if $\overrightarrow{x_j}$ has negative entries, we adopt a numerical stability technique to ensure we get a valid probability distribution even when $\overrightarrow{x_j}$ contains negative entries. Let

$$M_j := - \min_{\substack{v \in [n] \\ (\overrightarrow{x_j})_v \leq 0}} (\overrightarrow{x_j})_v \tag{25}$$

be the absolute value of the most negative entry of $\overrightarrow{x_j}$. If $\overrightarrow{x_j} \geq 0$, then set $M_j = 0$. Now, if $\overrightarrow{M_j} := M_j \cdot 1^n \in \mathbb{R}^{n \times 1}$, then $\overrightarrow{x_j'} = \overrightarrow{x_j} + \overrightarrow{M_j} \geq 0$. Therefore, we can estimate

$$\widehat{p'}_j \approx \overrightarrow{\pi_j'} := P^T \cdot \overrightarrow{x_j'} \tag{26}$$

using our Markov Chain method. Going back to our original goal of estimating $\overrightarrow{\pi}_j$, we have:

$$\overrightarrow{\pi_j} := P^T \cdot \overrightarrow{x_j} = P^T \cdot (\overrightarrow{x_j'} - \overrightarrow{M_j}) \tag{27}$$
$$= \overrightarrow{\pi_j'} - P^T \cdot \overrightarrow{M_j} = \overrightarrow{\pi_j'} - M_j \cdot P^T \cdot 1^n \tag{28}$$

where $1^n$ is the all 1-s vector. So then we only need to additionally estimate $P^T \cdot 1^n$. This can be done in the same fashion only once, as a pre-processing step. Specifically, suppose that we estimate $P^T \cdot 1^n$ as $\widehat{s}$. We know from our prior analysis that using $\Theta(\varepsilon^{-2}\log n)$ random walks we get an estimate $\widehat{s}$ such that:

$$\left|\left|\widehat{s} - P^T \cdot 1^n\right|\right|_\infty \leq \varepsilon n \tag{29}$$

Putting it all together, our final estimator is then:

$$\widehat{p}_j := \widehat{p'}_j - M_j \cdot \widehat{s} \tag{30}$$

Eventually, the total error for estimating $\overrightarrow{\pi_j}$ becomes:

$$\left|\left|\widehat{D}_{:,j}^V - D_{:,j}^V\right|\right|_\infty = ||\widehat{p}_j - \overrightarrow{\pi_j}||_\infty = \left|\left|\widehat{p'}_j - M_j \cdot \widehat{s} - \overrightarrow{\pi_j'} + M_j \cdot P^T \cdot 1^n\right|\right|_\infty \tag{31}$$
$$\leq \left|\left|\widehat{p'}_j - \overrightarrow{\pi_j'}\right|\right|_\infty + M_j \cdot \left|\left|P^T 1^n - \widehat{s}\right|\right|_\infty \tag{32}$$
$$\leq \varepsilon\left|\left|\overrightarrow{x_j'}\right|\right|_1 + \varepsilon M_j \cdot n \tag{33}$$
$$= \varepsilon\langle x_j, 1^n\rangle + 2\varepsilon n M_j \tag{34}$$

where the first inequality follows from the triangle inequality and the last equality follows from

$$\left|\left|\overrightarrow{x_j'}\right|\right|_1 = \sum_{k=1}^n |(x_j)_k + M_j| = \sum_{k=1}^n [(x_j)_k + M_j] = \langle x_j, 1^n\rangle + n M_j \tag{35}$$

---

**Algorithm 3** Estimating $P^T x$, with query access to $P \in \mathbb{R}^{n \times n}$ a stochastic matrix

---

1: **procedure** APPROXPOSPROD($P \in \mathbb{R}^{n \times n}, x \geq 0, \varepsilon > 0$)
2:      Let $N \leftarrow 2 \log n \cdot \varepsilon^{-2}$ and $\Sigma \leftarrow \langle x, 1^n \rangle$             $\triangleright O(n)$ time
3:      Let $\widehat{x} \in \mathbb{R}^{n \times 1}$ be our output.
4:      **for** $s \in [N]$ **do**
5:          Sample $i \in [n]$ with probability $\propto x_i$ using $\Sigma$ as a normalization factor.
6:          Sample $k \in [n]$ with probability $P_{ik}$.
7:          $\widehat{x}_k \leftarrow \widehat{x}_k + 1$
8:      **return** $\frac{1}{N} \cdot \widehat{x} \cdot \Sigma$
9: **procedure** ESTIMATEPRODUCT($P \in \mathbb{R}^{n \times n}, x \in \mathbb{R}^n, \varepsilon > 0, \widehat{s} \in \mathbb{R}^n$)
10:      Let $M \leftarrow -\min_{v \in [n], x_v \leq 0} x_v$             $\triangleright O(n)$ time
11:      Let $x' \leftarrow x + M \cdot 1^n$             $\triangleright O(n)$ time
12:      Call APPROXPOSPROD($P, x', \varepsilon$) to get $\widehat{x'}$             $\triangleright \widetilde{O}(n\varepsilon^{-2})$ time
13:      **return** $\widehat{x'} - M \cdot \widehat{s}$

---

**Algorithm 4** Estimating $D^V$

---

1: **Input:** $Q, K, D^O \in \mathbb{R}^{n \times d}$, error parameter $\varepsilon > 0$
2: Let $\widehat{D}^V \in \mathbb{R}^{n \times d}$ be our output.
3: $\widehat{s} \leftarrow$ APPROXPOSPROD($P, 1^n, \varepsilon$)             $\triangleright$ Pre-Processing
4: **for** $j \in [d]$ **do**
5:      $\widehat{D}^V_{:,j} \leftarrow$ ESTIMATEPRODUCT($P, D^O_{:,j}, \varepsilon, \widehat{s}$)
6: **return** $\widehat{D}^V$

---

**Runtime analysis**    For each $j \in [d]$ we take $N = O(\varepsilon^{-2} \log n)$ samples. We can take one sample in $O(nd)$ time. In addition, we must pre-calculate the sums $\langle x_j, 1^n \rangle + n M_j = n M_j + \sum_{k=1}^n D^O_{kj}$ for all $j \in [d]$, which takes $O(nd)$ time.      $\square$

**Remark 3.1.** *Note that Algorithm 4 does not materialize the $P$ matrix. Instead it accesses its elements by using $Q$ and $K$ in $O(d)$ time per element.*

## 4 EXPERIMENTAL RESULTS

In this section we present our experimental results. Through them we can interpret our theoretical framework better and solidify our understanding of it.

### 4.1 FORWARD PASS APPROXIMATION QUALITY ON RANDOM INPUTS

We begin by evaluating the effectiveness of $k$NN Attention in approximating the attention function. We randomly sample matrices $Q, K, V \in \mathbb{R}^{n \times d}$ from a uniform distribution over $[-B, B]^{n \times d}$ and assess the approximation quality on these inputs. Our focus is on the $k$NN Attention estimator we develop in Theorem 2.5 with $\lambda = 1$, as used in implementations like Bertsch et al. (2024) and Wu et al. (2022). We vary $k$ to study how the *absolute* error decreases as $k$ increases and compare the efficiency to the naive $O(n^2)$ attention, expecting notable performance gains. This experiment is implemented in PyTorch, running on a MacBook Air with an M3 CPU and 8GB of RAM.

**Efficiency of kNN Attention**    Our experiments confirm $k$NN Attention's superior speed, demonstrating sub-quadratic scaling. With a batch size of 1 and $H = 10$ attention heads, it handles self-attention for $n = 10^6$, while the naive method runs out of memory beyond $n \geq 20000$. Increasing $k$ further leads to memory errors for $n \geq 50000$, highlighting $k$NN Attention's memory efficiency. Detailed results are in Figure 2(a).

**Role of $k$ in the Approximation Error**    We investigate the impact of $k$ on the approximation error, predicting that error increases as $k$ decreases. The experiment confirms this, showing that for

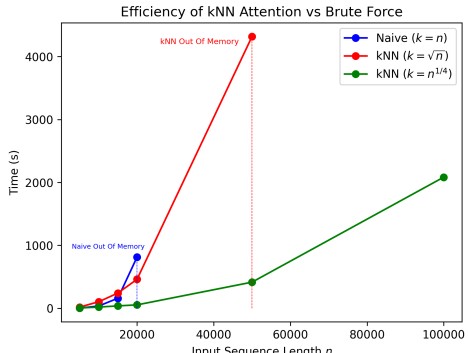
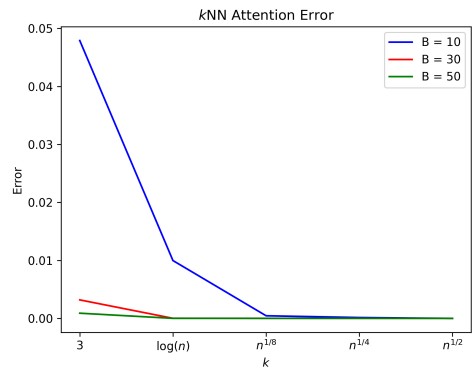

(a) $k$NN Attention vs Brute Force. We are able to increase the context length by a factor of 5 without running out of memory.

(b) Mean error of $k$NN Attention as a function of $k$ for different values of $B$. As $k$ grows, the error becomes negligible. In some cases, $\sqrt{n}$ is too big a threshold.

Figure 2: Experimental evaluation of the approximation quality and efficiency of $k$NN Attention on randomly generated input matrices $Q, K, V$.

$k \geq n^{1/8}$, the error is minimal. Our theory suggests a threshold closer to $\sqrt{n}$ which indicates that the optimal $k$ may vary by dataset. The results appear in Figure 2(b). Interestingly, the error is more pronounced for small values of both $B$ and $k$, potentially due to the limited approximation power when $k$ is small. For larger $k$, this difference becomes negligible.

## 4.2 BACKWARD PASS APPROXIMATION QUALITY

Next, we evaluate the quality of our algorithms for attention gradient estimation. We sample $Q, K, V$ from a normal distribution, as this strategy aligns with typical neural network weight initialization strategies, and approximate $D^Q$ and $D^V$ using randomized techniques. Our goal is to assess the error introduced by the approximation and whether this error causes gradient descent to converge far from the minimum.

We set the sequence length $N = 100$ and the embedding dimension $d = 3$. The learning rate $\alpha$ is varied between 0.05 and 0.5, while the error parameter is fixed at $\varepsilon = 0.05$ and the confidence parameter at $\delta = 0.1$. We experiment with both convex (Mean Square Error) and non-convex (Cross Entropy) loss functions to examine how approximate gradient descent behaves, using PyTorch's autograd to compute the exact attention gradients. As shown in Figure 3, our approximation closely matches the expected results in the convex case but deviates from the optimal convergence in the non-convex case. A more detailed investigation of the impact of gradient approximations in large language model (LLM) training is left for future work.

## 4.3 EXPERIMENTS ON LLMs

Finally, we experiment with incorporating $k$NN attention into LLMs to study its impact on training and inference. While previous work has explored $k$NN indices for efficient fine-tuning and training (Bertsch et al., 2024; Wu et al., 2022), our goal is to understand how Transformer LLMs respond to attention function approximation, linking it to our theory and providing practical guidelines. The architecture and training methods are adapted from *nanoGPT* (Karpathy, 2022), and our experiments are conducted on an NVIDIA L40 GPU with 48GB of memory.

Our first experiment trains a mini character-level Transformer on a small Shakespeare dataset, replacing attention with $k$NN attention. The sequence length is set to $N = 1024$, the embedding dimension to $d = 768$, learning rate to 0.005 and we perform $T = 5000$ iterations. We compare training and validation perplexity to the exact method. Results in Figure 4 show that $k$NN Attention maintains a small perplexity gap. However, as overfitting occurs, the perplexity difference widens, possibly due to increasing maximum approximation error. Future work could explore the impact of larger $k$ values on perplexity.

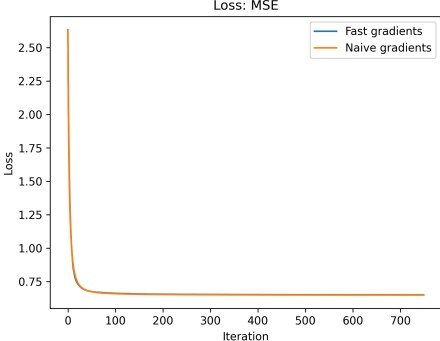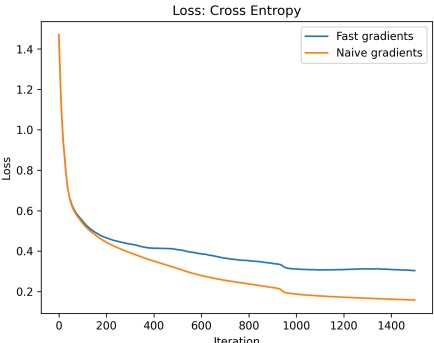

Figure 3: Gradient Descent with Approximate Gradients against different loss functions $\phi$. Even with approximate gradients, gradient descent still makes adequate progress towards convergence.

We also experiment with fine-tuning a large pre-trained LLM using $k$NN attention, in the hopes that the approximation will not severely degrade the model's quality. We manage to fine-tune GPT-2 XL[4] on a Shakespeare dataset - a task typically infeasible on a single L40 GPU due to memory constraints with quadratic attention. Examples from prompting this model can be found in Appendix H.

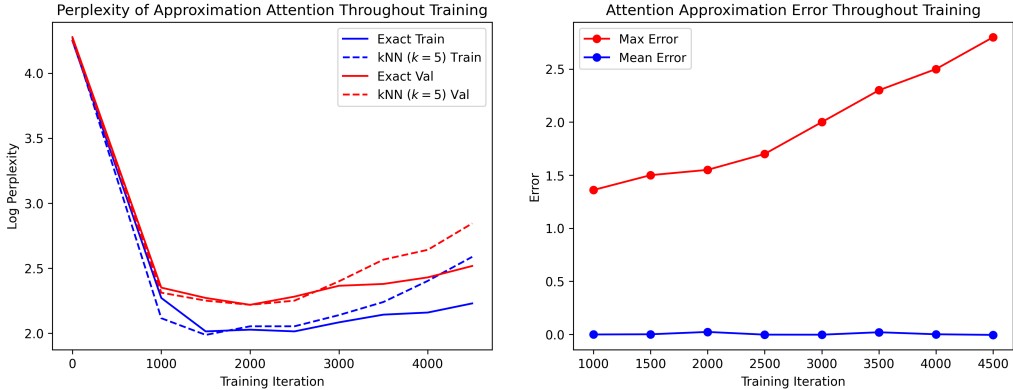

Figure 4: The perplexity and approximation error of $k$NN Attention throughout training

## 5 CONCLUSION

In this work, we developed a theoretical framework for leveraging $k$NN techniques to design efficient and effective Transformer architectures. We extended this framework by introducing Markov Chain-based methods to propose novel algorithms for efficient self-attention gradient computation. Empirical validation on both synthetic inputs and real-world datasets demonstrated that $k$NN approximations closely match the original performance in LLM training, while significantly reducing computational costs during both training and inference.

Moreover, our research opens several avenues for future exploration. Key questions include the effectiveness of training with approximate gradients compared to exact ones, particularly when error distributions are tightly concentrated but unknown. Another open question is about explaining the practical observation that the optimal $k$ value is often significantly smaller than the predicted $\sqrt{n}$. Overall, an improved understanding of the theoretical guarantees of $k$NN Attention can help solidify its place among other sparse attention techniques that provably attain close to linear time, while maintaining approximation quality.

In conclusion, by applying sublinear algorithm techniques, our work provides a solid foundation for making Transformers more scalable and efficient while identifying critical areas for further research in LLM approximation.

---

[4]1.61B parameters, see Radford et al. (2019)

## REPRODUCIBILITY STATEMENT

To aid in the reproduction of our experiments, we include our code in the repository at https://github.com/sansui-123/knn_attention. For our theoretical contributions, full proofs of our claims and analyses of our algorithms can be found in the Appendix.

## ACKNOWLEDGEMENTS

We thank Andrea Lincoln, Thien Nguyen, and Krzysztof Onak for valuable discussions at the early stages of this project. We are also grateful to Brian Kulis and Esty Kelman for their invaluable feedback on the manuscript and to Nikos Georgoudios for his assistance with the MGF proof of Lemma 2.1.

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

APPENDIX

In the following sections we deposit theoretical results, proofs and algorithms that are missing from the main paper, either due to space constraints or for the sake of clarity.

## A PRELIMINARIES

### A.1 SELF-ATTENTION AND APPROXIMATION

We start by defining self-attention and some of its variants.

**Definition A.1** (Self-Attention). *Let $Q, K, V \in \mathbb{R}^{n \times d}$. We can think of these matrices as a collection of $n$ $d$-dimensional query, key and value vectors respectively. We define the **self-attention** function as follows:*

$$O(Q, K, V) = D^{-1}AV \tag{36}$$

*where $A = \exp(QK^T) \in \mathbb{R}^{n \times n}$ is the **attention matrix** and $D = diag(A1^n)$. D effectively implements taking a row-wise softmax of the entries of $QK^T$. Many modern implementations of attention consider **causal attention**, in which we mask away the upper-triangular entries of A.*

**Remark A.1** (Normalization by $\sqrt{d}$). *In most implementations we divide $QK^T$ by $\sqrt{d}$ (Vaswani, 2017) because it reduces the variance of each element of A had $Q, K, T$ been selected from a uniform distribution. We will omit this technicality because it does not affect our algorithmic techniques. For the rest of this paper we will assume that $d^{-1/2}$ has been pre-normalized into $K$.*

**Remark A.2** (Dropout). *Many attention implementations also use **dropout**. Every entry of A will be masked to $0$ with probability $p$, where $p$ is set to a small constant, like $0.1$.*

**Definition A.2** (($\varepsilon, \delta$)-estimators). *Let $X$ be a statistic and $\widehat{X}$ be an estimator we have for it. $\widehat{X}$ is an $(\varepsilon, \delta)-$**additive estimator** if with probability at least $1 - \delta$ it holds that*

$$\left| X - \widehat{X} \right| \leq \varepsilon$$

*Respectively, we call the estimator **multiplicative** if*

$$\left| X - \widehat{X} \right| \leq \varepsilon X$$

We will often make use of the following boosting lemma from the theory of randomized algorithms:

**Lemma A.1** (Median-Of-Means Amplification Technique, (Chakrabarti, 2020)). *If $\widehat{Q}$ is an unbiased estimator of some statistic, then one can obtain an $(\varepsilon, \delta)$-multiplicative estimate of that statistic by suitably combining*

$$K := \frac{C}{\varepsilon^2} \ln \frac{2}{\delta} \frac{Var[\widehat{Q}]}{\mathbb{E}[\widehat{Q}]^2}$$

*independent samples of $\widehat{Q}$, where $C$ is a universal constant.*

### A.2 GUMBEL NOISE

The **Gumbel Distribution** will be useful for sampling from softmax distributions. We define it below:

**Definition A.3** (Gumbel Distribution). *The Gumbel distribution with mean $\mu$ and parameter $\beta$ has the following probability density function:*

$$Gumbel(\mu, \beta)(x) = \frac{1}{\beta} e^{-e^{-(x-\mu)/\beta}} \tag{37}$$

We will make use of the following properties of the Gumbel Distribution:

**Lemma A.2** (Gumbel Distribution Properties, (Chattamvelli & Shanmugam, 2021)). *The following are true:*

- *The mean of a Gumbel distribution is $\mu + \beta\gamma$[5]*

- *The moment generating function (MGF) of the Gumbel$(\mu, \beta)$ distribution is:*

$$M(t) = \Gamma(1 - \beta t)e^{\mu t} \tag{38}$$

  *where $\Gamma$ is the Gamma function.*

- *We can easily sample a Gumbel random variable of mean $\mu$ and parameter $\beta$ by using the uniform distribution:*

$$X = \mu - \beta\ln(-\ln(U)), \quad \text{where } U \sim \text{unif}([0, 1]) \tag{39}$$

- *Let $M_1, ..., M_n$ be $(\mu, \beta)$ independent Gumbel random variables. Then,*

$$M := \max_{i \in [n]} M_i$$

  *is a $(\mu + \beta\ln n, \beta)$ Gumbel random variable.*

Next, we present the well-known **Gumbel Max Trick**, an alternative way to sample from a softmax distribution:

**Lemma A.3** (Gumbel-Max-Trick, (Huijben et al., 2022)). *Let $x_1, ...x_n$ be real numbers and consider the softmax categorical distribution $p$ where*

$$p_i = \frac{\exp(x_i)}{\sum\limits_{k=1}^{n} \exp(x_i)}$$

*Consider sampling $n$ Gumbel random variables $G_1, ..., G_n \sim \text{Gumbel}(0, 1)$ and let*

$$\widehat{i} \in \arg\max_{i \in [n]}\{x_i + G_i\}$$

*Then $\widehat{i}$ is distributed according to $p$.*

**Binomial Distribution**    As part of our notational conventions, we also let $X \sim \text{Bin}(n, p)$ be a random variable distributed according to the Binomial Distribution with parameters $n$ and $p$.

## A.3    LOCALITY SENSITIVE HASHING

In our theoretical exposition we will make extensive use of schemes for *approximate nearest neighbor search*. A very successful such suite of algorithms with a long history (Andoni et al., 2014; 2015) of provable theoretical guarantees is Locality Sensitive Hashing (LSH):

**Theorem A.1** (Existence of LSH, (Gionis et al., 1999; Mussmann et al., 2017)). *Let $V \subseteq U$ be a set of size $n$ with a similarity measure $Sim(\cdot, \cdot)$. Consider a hash family $H$ such that for scalars $S_1 > S_2$ and $p_1 > p_2$:*

- *For any $x, y \in V$ where $Sim(x, y) \geq S_1$, $\Pr_{h \in H}[h(x) = h(y)] \geq p_1$*

- *For any $x, y \in V$ where $Sim(x, y) \leq S_2$, $\Pr_{h \in H}[h(x) = h(y)] \leq p_2$*

*This is called an $(S_1, S_2, p_1, p_2)$-**Locality Sensitive Hash Family**. Given such a family, one can construct a data structure which, given any query $q \in U$, does the following with high probability: if there exists some point $v \in V$ with $Sim(v, q) \geq S_1$, it returns a point $v' \in V$ with $Sim(v', q) \geq S_2$. If no point $v \in V$ exists with $Sim(v, q) \geq S_2$, it returns a negative answer $\perp$. Further, this can be done with $\widetilde{O}(n^\rho)$ query time and $\widetilde{O}(n^{1+\rho})$ space where $\rho = \log p_1 / \log p_2 < 1$.*

LSH also finds numerous applications in solving the Maximum Inner Product Search Problem (MIPS) as shown in Neyshabur & Srebro (2015), Shrivastava & Li (2014), and others.

---

[5]$\gamma \approx 0.577$ is the Euler-Mascheroni constant.

## A.4 Random Walks and some concentration bounds

For our back-propagation algorithms we will make use of some elementary tools from the theory of Random Walks.

**Definition A.4** (Random Walks). *Consider a state space $V = [n]$ and a weighted complete graph on $V$ with weights $w$ in $[0, 1]$ such that for all $u \in V$*

$$\sum_{v \in V} w_{uv} = 1$$

*This graph represents a **random walk** with transition matrix $P \in [0, 1]^{n \times n}$, where $P_{ij} = w_{ij}$. $P$ is a stochastic matrix because its rows sum to 1. In a random walk, we start at some vertex and choose a neighbor to jump to according to the probability distribution in $P$. The choice at each vertex conditioned on the previous transitions only depends on the vertex itself. This is known as the Markov Property.*

A first elementary observation is that if we start with a distribution $p$ over $V$ and we do a single step in the random walk, we can obtain the resulting distribution by multiplying the original distribution with $P^T$:

**Lemma A.4** (Single Step Random Walk Transition). *Consider a distribution $p \in \Delta(n)$[6] over $V$. Then, the quantity $P^T \cdot p$ gives the distribution over $V$ after one step of the random walk.*

*Proof.* Let $q$ be the distribution after one step. Let $v \in V$. We have by law of total probability that:

$$q(v) = \sum_{u \in V} p(u) \cdot w_{uv} = (P^T p)_v$$

$\square$

Finally, we state a well-known concentration result about independent random variables:

**Lemma A.5** (Hoeffding Bound). *Let $X_1, ..., X_n$ be independent random variables where $a_i \leq X_i \leq b_i$ almost surely. Let $S_n := X_1 + \cdots + X_n$. We have that:*

$$\Pr\left[|S_n - \mathbb{E}[S_n]| \geq t\right] \leq 2 \exp\left(-\frac{2t^2}{\sum_{i=1}^{n}(b_i - a_i)^2}\right)$$

## B Proof of Lemma 2.1

We prove Lemma 2.1. Our proof deviates from the proof of Mussmann et al. (2017) in that it uses a MGF-based argument, which we believe is cleaner.

**Lemma B.1** (Reminder). *In the context of Algorithm 1, we have that:*

$$\mathbb{E}[m] \leq \frac{n}{k} \tag{40}$$

*Proof.* By Lemma A.2, we can generate Gumbel$(0, 1)$ random variables as follows: Let $U_j$ be uniform in $[0, 1]$. Then:

$$G_{ij} = -\ln(-\ln(U_j)) \tag{41}$$

is distributed according to Gumbel$(0, 1)$. We want $G_{ij} > B$, which implies that:

$$-\ln(-\ln(U_j)) > B \iff U_j > \exp(-\exp(-B)) \tag{42}$$

---

[6]$\Delta(n) := \{x \in \mathbb{R}^n \mid x \geq 0, ||x||_1 = 1\}$ is the probability simplex over $[n]$.

So the number of points for which the Gumbel noise exceeds $B$ is distributed according to the Binomial distribution with parameters $n - k$ and $1 - \exp(-\exp(-B))$. If we condition on $M := \max_{j \in S_i}\{q_i^T k_j + G_{ij}\}$, we have that:

$$\mathbb{E}[m \mid M] = (n - k)(1 - \exp(-\exp(-B))) \tag{43}$$
$$\leq n \exp(-B) \tag{44}$$

where the last inequality follows by $e^{-x} \geq 1 - x$:

$$1 - \exp(-\exp(-B)) \leq 1 - (1 - \exp(-B)) = \exp(-B)$$

Now we can bound $\mathbb{E}[n \exp(-B)]$ by using the MGF of the Gumbel distribution (see Lemma A.2). Let $M' := \max_{j \in S_i} G_{ij}$. Recall by Lemma A.2 that $M'$ is a Gumbel random variable with $\mu_{M'} = \log k$ and $\beta_{M'} = 1$. Let $f_{M'}(t) = \mathbb{E}[e^{tM'}]$ be its moment generating function. We know that:

$$f_{M'}(t) = \Gamma(1 - t) \cdot e^{(\log k)t} = \Gamma(1 - t) \cdot k^t \tag{45}$$

This allows us to write:

$$\mathbb{E}[n \exp(-B)] = n \cdot \mathbb{E}[\exp(-B)] \tag{46}$$
$$= n \cdot \mathbb{E}[\exp(S_{\min} - M)] \tag{47}$$
$$= n \cdot \mathbb{E}[\exp(S_{\min} - \max_{j \in S_i}\{Z_{ij} + G_{ij}\})] \tag{48}$$
$$\leq n \cdot \mathbb{E}[\exp(S_{\min} - \min_{j \in S_i} Z_{ij} - M'] \tag{49}$$
$$= n \cdot \mathbb{E}[\exp(-M')] \tag{50}$$
$$= n \cdot f_{M'}(-1) \tag{51}$$
$$= \frac{n}{k} \tag{52}$$

where inequality 49 follows because

$$\max_{j \in S_i}\{Z_{ij} + G_{ij}\} \geq \min_{j \in S_i}\{Z_{ij}\} + \max_{j \in S_i} G_{ij} = S_{\min} + M'$$

For a quick proof of this statement, let $\widehat{j} := \arg\min_{j \in S_i} Z_{ij}$ and $\widetilde{j} := \arg\max_{j \in S_i} G_{ij}$. Also let $j^* := \arg\max_{j \in S_i}\{Z_{ij} + G_{ij}\}$. Then we have:

$$\max_{j \in S_i}\{Z_{ij} + G_{ij}\} = Z_{ij^*} + G_{ij^*} \tag{53}$$
$$\geq Z_{i\widetilde{j}} + G_{i\widetilde{j}} \tag{54}$$
$$\geq Z_{i\widehat{j}} + G_{i\widetilde{j}} \tag{55}$$
$$= S_{\min} + M' \tag{56}$$

Finally 52 follows because $\Gamma(2) = 2! = 1$. Now, via law of total expectation we finally get:

$$\mathbb{E}[m] = \mathbb{E}_M\left[\mathbb{E}[m \mid M]\right] \leq \mathbb{E}_M\left[\frac{n}{k}\right] = \frac{n}{k} \tag{57}$$

$\square$

## C $k$NN ATTENTION VIA CONCENTRIC LSH

In the main paper, we abstracted away the specific $k$NN method used to obtain the top-$k$ key vectors $k_j$ for every query vector $q_i$. In this section we cover a method for solving the $k$-MIPS problem in sub-linear time per query that has sound theoretical guarantees. In the context of Algorithm 2, this method could substitute the $k$NN index $H$.

This approach in question was proposed by Mussmann et al. (2017) and it uses a concentric LSH construction to get an approximation to this problem. First, let us define the approximate version of the $k$-MIPS problem, as is proposed in Mussmann et al. (2017):

**Definition C.1** (Approximate $k$-MIPS). *We say that a set $S_i$ is an approximate top-$k$ inner product solution if $|S_i| = k$ and there exists a constant $c$ such that:*

$$\max_{j \notin S_i} q_i^T k_j - \min_{j \in S_i} q_i^T k_j < c \tag{58}$$

If we are able to generate an approximate solution $S_i$ instead of an exact one, we have to lower our threshold $B$ to $M - S_{\min} - c$ in Algorithm 1, which in turns implies that $E[m] \leq \sqrt{n} \cdot e^c$. This remains sublinear in $n$ because $c$ is a constant.

The solution to the approximate version of the problem is constructed using LSH. Specifically, we build a sequence of $O(\text{polylog}(n, d))$ LSH data structures, each with concentric approximation radii, and hash all the key vectors $k_j \in \mathbb{R}^d$ into them. For each query vector $q_i$, we hash it across all these data structures and identify the first pair of consecutive LSH structures, $D_i$ and $D_{i+1}$, where $D_{i+1}$ contains more than $\sqrt{n}$ points in the buckets corresponding to $q_i$, while $D_i$ contains fewer than $\sqrt{n}$ points. For further details, see Mussmann et al. (2017). The following theorem ultimately holds:

**Theorem C.1** (Mussmann et al. (2017)). *Let $0 < \rho < 1$ be a constant. There exists an algorithm for solving the approximate version of $k$-MIPS on any single query $q$ with probability at least $1 - 1/n^2$ by using an explicit concentric LSH construction. The algorithm takes $O(dn^{1+\rho} \cdot \text{polylog}(n, d))$ pre-processing time/space, and $O(\sqrt{n} + n^\rho \cdot \text{polylog}(n, d))$ time/space per query.*

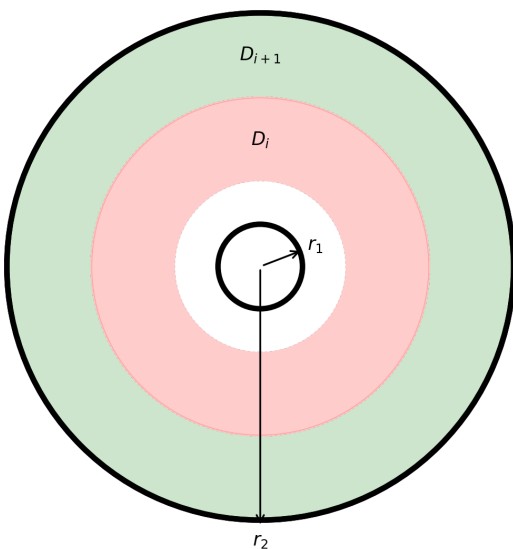

Figure 5: An illustration of the concentric LSH construction of Mussmann et al. (2017) In the $D_{i+1}$ band we find at least $\sqrt{n}$ points and in the $D_i$ band we find fewer than $\sqrt{n}$ points.

**Remark C.1** (The role of $\rho$). *The choice of $\rho < 1$ allows us to compute $S_i$ in sublinear time for each $i \in [n]$. The value of $\rho$ is determined by the radii gaps in the concentric construction. Our algorithm for computing $S_i$ is sublinear in $n$ because $\rho < 1$. Depending on the particular input dataset, we could have $\rho \leq 1/2$, which which case $f(n, \sqrt{n}) = \widetilde{O}(\sqrt{n})$ in the context of Theorem 6.*

## C.1   A COMPLETE ALGORITHM BASED ON SOLVING $k$-MIPS

We now have a complete algorithm to estimate self-attention with provable guarantees that is based on the solving $k$-MIPS problem for every query vector $q_i$. If we combine the boosted estimator approach of Theorem 2.1 with the Lazy Gumbel Sampling Technique of Algorithm 1 and the $k$-MIPS LSH technique of Theorem C.1, we arrive at the following theorem. We give pseudocode for the resulting algorithm, as Algorithm 5:

**Theorem C.2.** *Let $\varepsilon > 0$ and $\delta > 0$ be small positive constants. There exists an algorithm that can estimate Self-Attention in the same way as Theorem 2.1 and fail with probability at most $\delta + 1/n$. The algorithm's time complexity is shown in the table below, where $\rho \in (0, 1)$ is a fixed constant.*

|  | Pre-Processing | Main Computation |
|---|---|---|
| Complexity | $\widetilde{O}(dn^{1+\rho})$ | $\widetilde{O}\left(n^{1+\max\{1/2,\rho\}} \cdot d \cdot \varepsilon^{-2} \log(1/\delta)\right)$ |

---

**Algorithm 5** Approximating Self-Attention using concentric LSH $k$-MIPS solver

---

1: **Inputs:** $Q, K, V \in \mathbb{R}^{n \times d}$, error parameter $\varepsilon > 0$, confidence parameter $\delta > 0$

2: $H \leftarrow$ Create Concentric LSH data structures for solving $k$-MIPS, as in Mussmann et al. (2017)

3: Let $\widehat{O} \in \mathbb{R}^{n \times d}$ be our output.
4: **for** $i \in [n]$ **do**
5:    $S_i \leftarrow$ Query $H$ for the $\sqrt{n}$ indices $j \in [n]$ with the *approximate* largest values of $q_i^T k_j$
6:    **for** $j \in [d]$ **do**
7:       $\widehat{O_{ij}} \leftarrow$ Median-Of-Means with Algorithm 1 as sampler $\leftarrow (\sqrt{n}, q_i, K, S_i)$.
8: **return** $\widehat{O}$

---

*Proof.* Let $k = \sqrt{n}$. Suppose we construct the concentric LSH data structure according to Theorem C.1. This takes $\widetilde{O}(dn^{1+\rho})$ time. Let us condition on the event that for all queries $q_i$ the data structure provides a correct approximate answer $S_i$ to the $k$-MIPS problem. This happens with probability at least $1 - 1/n$ by union bound over all $n$ queries. Now we use our sets $S_i$ in Algorithm 1 to sample from $D_i$. Since retrieving $S_i$ takes $f(n, k) = O(\sqrt{n} + n^\rho \cdot \text{polylog}(n))$ time and space, Theorem 2.3 dictates that sampling from $D_i$ also takes $\widetilde{O}(n^{\max\{1/2,\rho\}})$ time and space. Thus, in the context of Theorem 2.1 we have that $T = \widetilde{O}(n^{\max\{1/2,\rho\}})$. Substituting back gives us the desired runtime and failure probability guarantees. $\qquad\square$

## D PROOF OF LEMMA 3.1: DERIVATION OF THE SELF-ATTENTION GRADIENTS

In this section, we will show the proof of Lemma 3.1. Suppose we have a differentiable scalar function $\phi$ that represents the loss when training our neural network after computing the output $O$: $\ell = \phi(O)$. Suppose that we have calculated $\frac{\partial \phi}{\partial O_{ij}}$ for all $i \in [n], j \in [d]$ and stored it in an matrix $D^O \in \mathbb{R}^{n \times d}$. Now we will calculate the remaining derivatives by using the chain rule. A similar calculation is also done in the Appendix of Dao et al. (2022).

**Calculating $\frac{\partial \phi}{\partial V_{ij}}$**    All these calculations just use the chain rule. One can simply draw a tree of dependencies and use it to perform the derivation. $\phi$ depends on $O_{ij}$ and $O_{ij}$ depends on all $V_{rj}$, so:

$$\frac{\partial \phi}{\partial V_{ij}} = \sum_{r=1}^{n} \frac{\partial \phi}{\partial O_{rj}} \cdot \frac{\partial O_{rj}}{\partial V_{ij}} = \sum_{r=1}^{n} D_{rj}^O \frac{\partial O_{rj}}{\partial V_{ij}}$$

Now, we calculate that:

$$\frac{\partial O_{rj}}{\partial V_{ij}} = \frac{\partial}{\partial V_{ij}} \sum_{k=1}^{n} P_{rk} V_{kj} = P_{ri}$$

so that gives:

$$\frac{\partial \phi}{\partial V_{ij}} = \sum_{r=1}^{n} D_{rj}^O P_{ri} = \sum_{r=1}^{n} P_{ir}^T D_{rj}^O \tag{59}$$

Thus, we can write the result succinctly:

$$D^V = P^T \cdot D^O \tag{60}$$

**Calculating** $\frac{\partial \phi}{\partial Q_{ij}}$    To do this, we will first calculate $\frac{\partial \phi}{\partial P_{ij}}$ and $\frac{\partial \phi}{\partial S_{ij}}$, where $S = QK^T$.

- First, each $O_{ij}$ depends on all $P_{ik}$, so the chain rule gives:

$$\frac{\partial \phi}{\partial P_{ij}} = \sum_{k=1}^{d} \frac{\partial \phi}{\partial O_{ik}} \cdot \frac{\partial O_{ik}}{\partial P_{ij}} = \sum_{k=1}^{d} D_{ik}^{O} \frac{\partial O_{ik}}{\partial P_{ij}}$$

We can calculate that:

$$\frac{\partial O_{ik}}{\partial P_{ij}} = V_{jk}$$

and so:

$$D_{ij}^{P} = \frac{\partial \phi}{\partial P_{ij}} = \sum_{k=1}^{d} D_{ik}^{O} V_{jk} = \langle D_{i,:}^{O}, V_{j,:} \rangle \tag{61}$$

for all $i \in [n], j \in [n]$.

- Now recall that $P_{ij} = \frac{\exp(S_{ij})}{L_i}$, so $P_{ij}$ depends on all $S_{ik}$ for $k = 1, ..., n$. Thus:

$$\frac{\partial \phi}{\partial S_{ij}} = \sum_{k=1}^{n} \frac{\partial \phi}{\partial P_{ik}} \cdot \frac{\partial P_{ik}}{\partial S_{ij}} = \sum_{k=1}^{n} D_{ik}^{P} \cdot \frac{\partial P_{ik}}{\partial S_{ij}}$$

$$= D_{ij}^{P} \cdot \frac{\partial P_{ij}}{\partial S_{ij}} + \sum_{k=1, k \neq j}^{n} D_{ik}^{P} \cdot \frac{\partial P_{ik}}{\partial S_{ij}}$$

We now calculate seperately the two cases by using the quotient rule:

- $k \neq j$:

$$\frac{\partial P_{ik}}{\partial S_{ij}} = \frac{\partial}{\partial S_{ij}} \frac{\exp(S_{ik})}{\sum\limits_{r=1}^{n} \exp(S_{ir})} = -\exp(S_{ik}) \cdot \frac{\exp(S_{ij})}{\left(\sum\limits_{r=1}^{n} \exp(S_{ir})\right)^2}$$

$$= -P_{ik}P_{ij}$$

- $k = j$:

$$\frac{\partial P_{ij}}{\partial S_{ij}} = \frac{\partial}{\partial S_{ij}} \frac{\exp(S_{ij})}{\sum\limits_{r=1}^{n} \exp(S_{ir})} = \frac{\exp(S_{ij}) \sum\limits_{r=1}^{n} \exp(S_{ir}) - \exp(S_{ij}) \exp(S_{ij})}{\left(\sum\limits_{r=1}^{n} \exp(S_{ir})\right)^2}$$

$$= P_{ij} - P_{ij}^2$$

Now we can put it all together:

$$\frac{\partial \phi}{\partial S_{ij}} = D_{ij}^{P} \cdot \frac{\partial P_{ij}}{\partial S_{ij}} + \sum_{k=1, k \neq j}^{n} D_{ik}^{P} \cdot \frac{\partial P_{ik}}{\partial S_{ij}}$$

$$= D_{ij}^{P} \cdot (P_{ij} - P_{ij}^2) - \sum_{k=1, k \neq j}^{n} D_{ik}^{P} \cdot P_{ik}P_{ij}$$

$$= D_{ij}^{P} \cdot P_{ij} - \sum_{k=1}^{n} D_{ik}^{P} \cdot P_{ik}P_{ij}$$

$$= P_{ij} \left( D_{ij}^{P} - \langle D_{i,:}^{P}, P_{i,:} \rangle \right)$$

Now finally, for $i \in [n], j \in [d]$, $Q_{ij}$ influences $S_{ik}$ for all $k \in [n]$, so:

$$\frac{\partial \phi}{\partial Q_{ij}} = \sum_{k=1}^{n} \frac{\partial \phi}{\partial S_{ik}} \frac{\partial S_{ik}}{\partial Q_{ij}} \tag{62}$$

$$= \sum_{k=1}^{n} P_{ik} \left( D_{ik}^{P} - \langle D_{i,:}^{P}, P_{i,:} \rangle \right) K_{kj} \tag{63}$$

**Calculating** $\frac{\partial \phi}{\partial K_{ij}}$   We know that $K_{ij}$ influences $S_{ki}$ for $k \in [n]$, so:

$$\frac{\partial \phi}{\partial K_{ij}} = \sum_{k=1}^{n} \frac{\partial \phi}{\partial S_{ki}} \frac{\partial S_{ki}}{\partial K_{ij}} \tag{64}$$

$$= \sum_{k=1}^{n} P_{ki} \left( D_{ki}^{P} - \langle D_{k,:}^{P}, P_{k,:} \rangle \right) Q_{kj} \tag{65}$$

## E   ESTIMATING $D^Q$

In this section we give an efficient algorithm for estimating $D^Q$. This algorithm is based on our $k$NN-Attention framework. Recall that we found that:

$$D_{ij}^{Q} = \sum_{k=1}^{n} P_{ik} \left( D_{ik}^{P} - \langle D_{i,:}^{P}, P_{i,:} \rangle \right) K_{kj}$$

We can write this expression as an expectation with respect to the distribution $D_i$:

$$\frac{\partial \phi}{\partial Q_{ij}} = \mathbb{E}_{k \sim D_i} \left[ D_{ik}^{P} K_{kj} \right] - \mathbb{E}_{k \sim D_i} \left[ K_{kj} \cdot \mathbb{E}_{s \sim D_i} [D_{is}^{P}] \right] \tag{66}$$

$$= \underbrace{\mathbb{E}_{k \sim D_i} \left[ D_{ik}^{P} K_{kj} \right]}_{E_1} - \underbrace{\mathbb{E}_{k \sim D_i} \left[ K_{kj} \right]}_{E_2} \cdot \underbrace{\mathbb{E}_{s \sim D_i} [D_{is}^{P}]}_{E_3} \tag{67}$$

This allows us to use any of our softmax expectation estimators. We choose the Median-Of-Means estimator for the purposes of a clean analysis. We just have to do it three times and ensure that the terms we take expectations over are efficiently computable. Indeed, because

$$D_{ik}^{P} = \langle D_{i,:}^{O}, V_{k,:} \rangle, \tag{68}$$

we can compute all three of those expectations in sublinear time! Let $\widehat{E_1}, \widehat{E_2}, \widehat{E_3}$ be the estimates we produce. Then, almost identically to the error analysis we did for the forward pass, we get an $(\varepsilon, \delta)$-*additive* estimate for $E_i$, where $i \in \{1, 2, 3\}$[7]

$$\Pr \left[ |\widehat{E_1} - E_1| \geq \varepsilon \right] \leq \frac{\delta}{3} \tag{69}$$

$$\Pr \left[ |\widehat{E_2} - E_2| \geq \varepsilon \right] \leq \frac{\delta}{3} \tag{70}$$

$$\Pr \left[ |\widehat{E_3} - E_3| \geq \varepsilon \right] \leq \frac{\delta}{3} \tag{71}$$

And so, putting these three together and using the union bound we get that with probability at least $1 - \delta$ it holds that:

$$\left| \widehat{E_1} - \widehat{E_2} \cdot \widehat{E_3} - E_1 + E_2 \cdot E_3 \right| \leq \left| \widehat{E_1} - E_1 \right| + \left| \widehat{E_2} \cdot \widehat{E_3} - E_2 \cdot E_3 \right| \tag{72}$$

$$= \left| \widehat{E_1} - E_1 \right| + \left| \widehat{E_2} \cdot \widehat{E_3} - \widehat{E_2} \cdot E_3 + \widehat{E_2} \cdot E_3 - E_2 \cdot E_3 \right| \tag{73}$$

$$\leq \left| \widehat{E_1} - E_1 \right| + \widehat{E_2} \left| \widehat{E_3} - E_3 \right| + E_3 \left| \widehat{E_2} - E_2 \right| \tag{74}$$

$$\leq \varepsilon + \varepsilon \cdot \widehat{E_2} + \varepsilon E_3 \tag{75}$$

$$\leq \varepsilon + \varepsilon (E_2 + \varepsilon) + \varepsilon E_3 \tag{76}$$

$$= \varepsilon + \varepsilon^2 + \varepsilon (E_2 + E_3) \tag{77}$$

In order to bound the variance of our estimators, we need to assume some bounds on the inputs, analogously to $||V||_\infty \leq B = O(\log n)$ from Theorem 2.1. First, we assume that $||K||_\infty \leq B_K =$

---

[7]In Theorem 2.1 we used a multiplicative approximation. To get the additive approximation guarantee we need $O(\varepsilon^{-2} \log(1/\delta) \cdot \text{Var}[\widehat{O}_{ij}])$ samples, where $\text{Var}[\widehat{O}_{ij}] \leq B^2 = O(\text{polylog}(n))$.

$O(\text{polylog}(n))$. Second, we have that $||D^P||_\infty \leq dB \cdot ||D^O||_\infty = O(\text{polylog}(n))$ if $d = O(\log n)$. This also gives that $||D^P \circ K||_\infty \leq B_K \cdot ||D^P||_\infty = O(\text{polylog}(n))$.

These assumptions are reasonable within the context of the hardness results proved for the attention mechanism and the computation of its gradients[8] (Alman & Song, 2024a;b). Given these assumptions, we can also bound the error more compactly. Starting from Equation 77, we get: $e_Q \leq O(\varepsilon) + \varepsilon(B_K + dB \cdot ||D^O||_\infty) = O(\varepsilon \cdot \text{polylog}(n))$. As a result, we arrive at the following theorem:

**Theorem E.1.** *Assume that* $||K||_\infty = O(polylog(n)), d = O(\log n)$ *and* $||D^O||_\infty = O(polylog(n))$. *There exists a sub-quadratic algorithm that takes as input* $Q, K, V, D^O \in \mathbb{R}^{n \times d}$ *and outputs a matrix* $\widehat{D}^Q \in \mathbb{R}^{n \times d}$ *such that:*

$$\left\|\widehat{D}^Q - D^Q\right\|_\infty \leq O(\varepsilon \cdot polylog(n)) \tag{78}$$

*This algorithm is shown as Algorithm 6.*

*Proof.* The proof and analysis of Algorithm 6 is detailed in the preceding paragraph. □

---

**Algorithm 6** Estimating $D^Q$

---

**procedure** ESTIMATE-$E_1(Q, K, V, D^O, S_i, i, j, \varepsilon, \delta)$
  $F \leftarrow \{\langle D^O_{i,:}, V_{k,:}\rangle \cdot K_{kj}\}^n_{k=1} \in \mathbb{R}^{n \times 1}$      ▷ *F will not be materialized.*
  $\widehat{E_1} \leftarrow$ Median-Of-Means with Lazy Gumbel Sampling $\leftarrow Q, K, F, S_i, \varepsilon, \delta$
  **return** $\widehat{E_1}$
**procedure** ESTIMATE-$E_2(Q, K, S_i, i, \varepsilon, \delta)$
  $\widehat{E_2} \leftarrow$ Median-Of-Means with Lazy Gumbel Sampling $\leftarrow Q, K, K_{:,j}, S_i, \varepsilon, \delta.$
  **return** $\widehat{E_2}.$
**procedure** ESTIMATE-$E_3(Q, K, V, D^O, S_i, i, \varepsilon, \delta)$
  $F \leftarrow \{\langle D^O_{i,:}, V_{k,:}\rangle\}^n_{k=1} \in \mathbb{R}^{n \times 1}$      ▷ *F will not be materialized.*
  $\widehat{E_3} \leftarrow$ Median-Of-Means with Lazy Gumbel Sampling $\leftarrow Q, K, F, S_i, \varepsilon, \delta$
  **return** $\widehat{E_3}$
**Input:** $D^O \in \mathbb{R}^{n \times d}, Q, K, V \in \mathbb{R}^{n \times d}$, parameters $\varepsilon, \delta > 0$
Let $\widehat{D}^Q \in \mathbb{R}^{n \times d}$ be our output.
**for** $i \in [n]$ **do**
  $S_i \leftarrow \sqrt{n}$ values $t \in [n]$ of the largest $q_i^T k_t$ via LSH or $k$NN.
  **for** $j \in [d]$ **do**
    $\widehat{E_1} \leftarrow$ ESTIMATE-$E_1(Q, K, V, D^O, S_i, i, j, \varepsilon, \delta)$
    $\widehat{E_2} \leftarrow$ ESTIMATE-$E_2(Q, K, S_i, i, \varepsilon, \delta)$
    $\widehat{E_3} \leftarrow$ ESTIMATE-$E_3(Q, K, V, D^O, S_i, i, \varepsilon, \delta)$
    $\widehat{D}^Q_{ij} \leftarrow \widehat{E_1} - \widehat{E_2} \cdot \widehat{E_3}$
**return** $\widehat{D}^Q$

---

# F  ESTIMATING $D^K$

Finally, we turn to estimating $D^K$. Our earlier calculations show that

$$\frac{\partial \phi}{\partial K_{ij}} = \sum_{k=1}^n P_{ki}\left(D^P_{ki} - \langle D^P_{k,:}, P_{k,:}\rangle\right) Q_{kj}$$

---

[8]Another motivation for assuming an upper bound on the norm of $D^O$ is to avoid the phenomenon of exploding gradients in training neural networks.

We can break up this sum into two terms:

$$\frac{\partial \phi}{\partial K_{ij}} = \underbrace{\sum_{k=1}^{n} P_{ki} D_{ki}^{P} Q_{kj}}_{A_{ij}} - \underbrace{\sum_{k=1}^{n} P_{ki} \langle D_{k,:}^{P}, P_{k,:} \rangle \cdot Q_{kj}}_{B_{ij}} \tag{79}$$

We will estimate both terms separately:

## F.1   ESTIMATING $A_{ij}$

For $i \in [n]$ and $j \in [d]$, we have:

$$A_{ij} = \sum_{k=1}^{n} P_{ki} D_{ki}^{P} Q_{kj} = \sum_{k=1}^{n} P_{ki} Q_{kj} \cdot \langle D_{k,:}^{O}, V_{i,:} \rangle \tag{80}$$

$$= \sum_{k=1}^{n} P_{ik}^{T} \cdot Y_{kj}^{(i)} \tag{81}$$

where $Y_{kj}^{(i)} := Q_{kj} \cdot \langle D_{k,:}^{O}, V_{i,:} \rangle$. So we can write:

$$A_{ij} = \underbrace{(P^{T})_{i,:}}_{1 \times n} \cdot \underbrace{Y_{:,j}^{(i)}}_{n \times 1} \tag{82}$$

We will use our familiar Markov Chain estimation method from Algorithm 3 to calculate this quantity. However, in this case we only care about estimating the $i$-th entry in the vector $(P^{T}) \cdot Y_{:,j}^{(i)}$, which we can do by performing $O(\log n \cdot \varepsilon^{-2})$ simulations. Ultimately, by following the same analysis as in Algorithm 3, we are able to estimate $A_{ij}$ with probability at least $1 - \frac{1}{n}$ and error:

$$\left| \widehat{A}_{ij} - A_{ij} \right| \leq \varepsilon \langle Y_{:,j}^{(i)}, 1^{n} \rangle + 2\varepsilon n M_{j}^{(i)} \tag{83}$$

$$= \varepsilon \sum_{k=1}^{n} Q_{kj} \cdot \langle D_{k,:}^{O}, V_{i,:} \rangle + 2\varepsilon n M_{j}^{(i)} \tag{84}$$

where

$$M_{j}^{(i)} = - \min_{\substack{k \in [n] \\ Y_{kj}^{(i)} \leq 0}} Y_{kj}^{(i)}$$

**Remark F.1.** *Because we would need to calculate all $n^2$ values of $M_{j}^{(i)}$, we will instead use a single upper bound $M \geq M_{j}^{(i)}$ for all $(i,j) \in [n] \times [d]$ for this algorithm. We assume that we know a large enough $M$ in advance and that $M = O(polylog(n))$.*

In the next paragraphs, we will tackle some implementation issues that arise in this approach. We did not see these issues when estimating $D^V$, and because they make the algorithm a lot more complicated, we left them for last.

**Pre-calculating the normalizing factors**   We need to pre-calculate the normalizing sums $N_{j}^{(i)} = \langle Y_{:,j}^{(i)}, 1^{n} \rangle + nM$ for all $(i,j) \in [n] \times [d]$. Naively, it takes $O(n^2 d)$ time to calculate all those sums. However, with some preprocessing we can take the time down to $O(nd^2)$. First, observe that we have:

$$N_{j}^{(i)} = nM + \langle Y_{:,j}^{(i)}, 1^{n} \rangle = nM + \sum_{k=1}^{n} Q_{kj} \cdot \langle D_{k,:}^{O}, V_{i,:} \rangle = nM + \langle V_{i,:}, \sum_{k=1}^{n} Q_{kj} \cdot D_{k,:}^{O} \rangle \tag{85}$$

We can thus first pre-compute the $d$ vectors $\overrightarrow{E_{j}} = \sum_{k=1}^{n} Q_{kj} \cdot D_{k,:}^{O} \in \mathbb{R}^d$ for each $j \in [d]$ in $O(nd^2)$ time. Then, for each $i \in [n]$ and $j \in [d]$, we can produce $N_{j}^{(i)}$ in $O(d)$ time by using Equation 85, bringing the total time complexity to $O(nd^2)$.

**Sampling according to $Y_{:,j}^{(i)} + M \cdot 1^n$ efficiently**    Unfortunately, because we are now estimating $A_{ij}$ individually for all $(i,j) \in [n] \times [d]$, we cannot spend $O(n)$ time to generate each sample. We need to generate samples in sublinear time with some pre-processing. This seems intuitively difficult at first because we have $O(nd)$ distributions over $[n]$ and each distribution requires $\Omega(n)$ time to sample one sample. However, we can take advantage of the structure between the distributions in order to reduce the pre-processing time. First, consider the following method of sampling from a distribution $[p_1, ..., p_n]$:

1. Compute the cumulative sums $s_i = \sum_{k=1}^{i} p_i$. We know that $s_1 = p_1$ and $s_n = 1$.

2. Pick some $x \sim \text{Unif}(0,1)$ uniformly at random from $(0,1)$.

3. Find the interval $[p_i, p_{i+1}]$ for $i \in [1, n-1]$ in which $x$ falls in. That is, find the smallest $i$ for which $x \leq s_i$. We can do this in $O(\log n)$ time using binary search.

4. Output $i$.

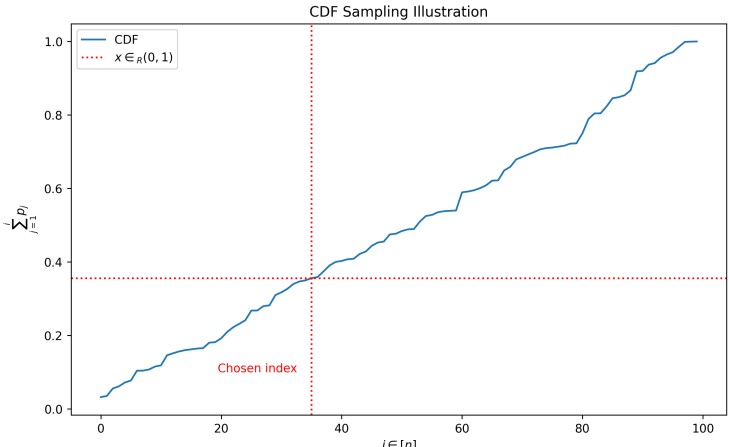

Figure 6: An illustration of the CDF sampling method: We form the CDF and then sample an index by choosing $x \in (0,1)$ and using binary search to find the corresponding bucket.

It is easy to see that this method outputs a value $i$ with probability $p_i$. If we applied this method naively we would still take $O(n^2 d)$ time because we'd have to calculate all the cumulative sums. However, the inner product structure again comes to our rescue:

$$Y_{kj}^{(i)} = \langle V_{i,:}, Q_{kj} \cdot D_{k,:}^O \rangle \tag{86}$$

So, we can create $d$ cumulative sum tables $\Sigma_j$ for $j \in [d]$, each of which stores $n$ cumulative-sum $\mathbb{R}^d$ vectors as follows:

$$(\Sigma_j)_\ell = \sum_{s=1}^{\ell} Q_{sj} \cdot D_{s,:}^O \in \mathbb{R}^d, \ \forall \ell \in [n] \tag{87}$$

This requires $O(nd^2)$ time and space to construct. Now, in order to sample with probability proportional to $Y_{kj}^{(i)} + M$ given that we know $N_j^{(i)}$, we sample $x_{ij} \sim \text{Unif}(0,1)$ and perform binary search to find the interval $x_{ij}$ belongs to. At that point, we can calculate the $O(\log n)$ necessary cumulative sums in $O(d)$ time each by using our pre-processing:

$$\sum_{s=1}^{\ell} \left( Y_{kj}^{(i)} + M \right) = kM + \langle V_{i,:}, (\Sigma_j)_\ell \rangle \tag{88}$$

This allows us to sample in $O(d \log n)$ time after a $O(nd^2)$ pre-processing. Our algorithm in total is included as part of Algorithm 7.

**Sampling with respect to** $D_i$    Again, we cannot afford to sample from the softmax naively with $O(n)$ time. Thankfully, we know of a sublinear method that can allow us to sample from the softmax, with slightly super-linear pre-processing time: the Lazy-Gumbel Sampling method. We will omit the pre-processing details in the algorithm pseudocode.

---

**Algorithm 7** Estimating $D^K$ – Part 1: Computing $A$

---

1: **Input:** $Q, K, V, D^O \in \mathbb{R}^{n \times d}$, error parameter $\varepsilon > 0$
2: **for** $j \in [d]$ **do**                                                ▷ Pre-Processing
3:      Compute $\overrightarrow{E_j} = \sum_{k=1}^{n} Q_{kj} \cdot D^O_{k,:} \in \mathbb{R}^d$
4:      Compute the cumulative sums $(\Sigma_j)_\ell = \sum_{s=1}^{\ell} Q_{sj} \cdot D^O_{s,:} \in \mathbb{R}^d$ for all $\ell \in [n]$.
5:      Compute $\widehat{s} \approx P^T 1^n$ using Markov Chain simulations.
6:      Initialize a $k$NN index $H$.
7: **procedure** COMPUTE$-A(Q, K, D^O, E, \Sigma, \varepsilon, H, \widehat{s}, M)$
8:      Let $N \leftarrow 2 \log n \cdot \varepsilon^{-2}$
9:      $\widehat{A} \leftarrow [0]^{n \times d}$ is the output.
10:      **for** $i \in [n]$ **do**
11:          Query $H$ to get set $S_i$
12:          **for** $j \in [d]$ **do**
13:              $N_j^{(i)} \leftarrow \langle V_{i,:}, \overrightarrow{E_j} \rangle + nM$                                  ▷ $O(d)$ time.
14:              **for** $s \in [N]$ **do**
15:                  Sample $k \in [n]$ with probability $\propto Y_{kj}^{(i)} + M$ via binary search, $\Sigma_j$ and $N_j^{(i)}$
16:                  Sample $\ell \in [n]$ with probability $P_{ik}$ via Lazy Gumbel Sampling, given $S_i$
17:                  **if** $\ell = i$ **then**
18:                      $\widehat{A}_{ij} \leftarrow \widehat{A}_{ij} + 1$
19:              $\widehat{A}_{ij} \leftarrow \frac{1}{N}(\widehat{A}_{ij} \cdot N_j^{(i)}) - M \cdot \widehat{s}_i$
20:      **return** $\widehat{A}$

---

### F.2    ESTIMATING $B_{ij}$

For $(i, j) \in [n] \times [d]$, we first have:

$$B_{ij} = \sum_{k=1}^{n} P_{ki} \cdot \langle D^P_{k,:}, P_{k,:} \rangle \cdot Q_{kj} \tag{89}$$

$$= \sum_{k=1}^{n} P_{ki} X_{kj} \tag{90}$$

where $X_{kj} = \langle D^P_{k,:}, P_{k,:} \rangle \cdot Q_{kj}$. Notice that $X_{kj}$ takes $O(nd)$ time to naively compute, so we will first approximate it with $\widehat{X}_{kj}$. Observe that:

$$X_{kj} = Q_{kj} \cdot \langle D^P_{k,:}, P_{k,:} \rangle \tag{91}$$

$$= Q_{kj} \cdot \sum_{s=1}^{n} D^P_{ks} \cdot P_{ks} \tag{92}$$

$$= Q_{kj} \cdot \mathbb{E}_{s \sim D_k}[D^P_{ks}] \tag{93}$$

$$= \mathbb{E}_{s \sim D_k}[Q_{kj} \cdot D^P_{ks}] \tag{94}$$

$$\approx \widehat{X}_{kj} \tag{95}$$

Let us approximate $\mathbb{E}_{s \sim D_k}[Q_{kj} \cdot (D^p)_{ks}] \approx \widehat{X}_{kj}$ using the Lazy Gumbel Sampling and Median-Of-Means method. This allows us to get for all $(k, j) \in [n] \times [d]$ with probability at least $1 - \delta$ that:

$$\left| \widehat{X}_{kj} - X_{kj} \right| \leq \varepsilon \tag{96}$$

**Remark F.2.** *To have an $o(n)$ bound for the variance, we have to assume (again) that $||X||_\infty = O(polylog(n))$. This follows from the assumption that $||Q||_\infty = O(polylog(n))$ and $||D^P||_\infty = O(polylog(n))$. The latter follows from $||D^O||_\infty = O(polylog(n))$. So the assumptions here are the same as in Theorem E.1.*

Now we can define:

$$\widehat{B}_{ij} = \sum_{k=1}^{n} P_{ki}\widehat{X}_{kj} \tag{97}$$

We can bound the error of this approximation using the triangle inequality:

$$\left|B_{ij} - \widehat{B}_{ij}\right| = \left|\sum_{k=1}^{n} P_{ki}(\widehat{X}_{kj} - X_{kj})\right| \tag{98}$$

$$\leq \sum_{k=1}^{n} P_{ki}\left|\widehat{X}_{kj} - X_{kj}\right| \tag{99}$$

$$\leq \varepsilon \sum_{k=1}^{n} P_{ki} \tag{100}$$

$$= \varepsilon\langle P_{:,i}, 1^n\rangle \tag{101}$$

Now the problem is calculating $\widehat{B}$. Note that we can write:

$$\widehat{B} = P^T \cdot \widehat{X} \tag{102}$$

Finally, this takes us back to the calculation of $D^V$. We can use the exact same Markov Chain method and get a final approximation $\widetilde{B}$ so that with probability at least $1 - \frac{1}{n}$ it holds that:

$$\left|\widetilde{B}_{ij} - \widehat{B}_{ij}\right| \leq \varepsilon\langle \widehat{X}_{:,j}, 1^n\rangle + 2\varepsilon n M_j^{(X)} \tag{103}$$

where

$$M_j^{(X)} := -\min_{\substack{k\in[n] \\ \widehat{X}_{kj}\leq 0}} \widehat{X}_{kj}$$

Then the overall error can be bounded as follows:

$$\left|\widetilde{B}_{ij} - B_{ij}\right| \leq \left|\widetilde{B}_{ij} - \widehat{B}_{ij}\right| + \left|\widehat{B}_{ij} - B_{ij}\right| \tag{104}$$

$$\leq \varepsilon\langle P_{:,i}, 1^n\rangle + \varepsilon\langle \widehat{X}_{:,j}, 1^n\rangle + 2\varepsilon n M_j^{(X)} \tag{105}$$

$$\leq \varepsilon\langle P_{:,i}, 1^n\rangle + \varepsilon\langle X_{:,j}, 1^n\rangle + \varepsilon^2 n + 2\varepsilon n M_j^{(X)} \tag{106}$$

$$= \varepsilon\langle P_{:,i} + X_{:,j}, 1^n\rangle + \varepsilon^2 n + 2\varepsilon n M_j^{(X)} \tag{107}$$

To wrap up our implementation details, we can calculate the required normalization sums as follows:

$$\langle \widehat{X}_{:,j}, 1^n\rangle = \sum_{k=1}^{n} \widehat{X}_{kj} = \sum_{k=1}^{n} Q_{kj}\widehat{D}_k \tag{108}$$

We can do this in $\approx \widetilde{O}(dn^{3/2})$ time if we precompute in advance

$$\widehat{D}_k := \langle D_{k,:}^P, P_{k,:}\rangle \tag{109}$$

using Lazy Gumbel Sampling for all $k \in [n]$. Further, each element $\widehat{X}_{ij}$ can be computed in $\approx \widetilde{O}(\sqrt{n})$ time as well in a similar fashion. Finally, $M_j^{(X)}$ can also be calculated in such time. Our algorithm is given below as Algorithm 8. By combining algorithms 7 and 8 we arrive at the following theorem for Algorithm 9:

---

**Algorithm 8** Estimating $D^K$ – Part 2: Computing $B$

---

1: $S_i \leftarrow$ Use an LSH or $k$NN index to calculate $S_i$ for all $i \in [n]$.
2: $\widehat{s} \leftarrow$ ESTIMATEPRODUCTPOSITIVE$(P, 1^n, \varepsilon)$
3: **procedure** COMPUTE$-\widehat{X}_{kj}(Q, K, D^O, V, S_i, \varepsilon, \delta, k, j)$
4:     $F \leftarrow \{Q_{kj} \cdot \langle D^O_{k,:}, V_{s,:} \rangle\}_{s=1}^n \in \mathbb{R}^{n \times 1}$     ▷ *F will not be materialized.*
5:     $\widehat{X}_{kj} \leftarrow$ Median-Of-Means with Lazy Gumbel Sampling $\leftarrow Q, K, F, S_i, \varepsilon, \delta$
6:     **return** $\widehat{X}_{kj}$.
7: **procedure** COMPUTE$-B(Q, K, V, D^O, \varepsilon)$
8:     Output $\widetilde{B} \leftarrow [0]^{n \times d}$
9:     **for** $j \in [d]$ **do**     ▷ $O(d)$ times
10:        $\widetilde{B}_{:,j} \leftarrow$ ESTIMATEPRODUCT$(P, \widehat{X}_{:,j}, \varepsilon, \widehat{s})$
11:    **return** $\widetilde{B}$

---

**Theorem F.1.** *There exists an algorithm that approximates $D^K$ on inputs $Q, K, V, D^O$ under our standard assumptions such that the estimate $\widehat{D}^K$ satisfies:*

$$\left\| \widehat{D}^K_{:,j} - D^K_{:,j} \right\|_\infty \leq \varepsilon \langle P_{:,i} + X_{:,j}, 1^n \rangle + \varepsilon^2 n + 2\varepsilon n M_j^{(X)}$$

$$+ \varepsilon \sum_{k=1}^n Q_{kj} \cdot \langle (D^o)_{k,:}, V_{i,:} \rangle + 2\varepsilon n M$$

*where:*

$$M \geq M_j^{(i)} := - \min_{\substack{k \in [n] \\ Y_{kj}^{(i)} \leq 0}} Y_{kj}^{(i)} \quad and \quad M_j^{(X)} := - \min_{\substack{k \in [n] \\ \widehat{X}_{kj} \leq 0}} \widehat{X}_{kj} \tag{110}$$

*under our previous definitions for all $j \in [d]$. The algorithm runs in sub-quadratic time and space and succeeds with probability $\geq 1 - \delta$.*

*Proof.* We have that $D^K_{ij} = A_{ij} - B_{ij}$. We define $\widehat{D}^K_{ij} = \widehat{A}_{ij} - \widetilde{B}_{ij}$ and we established in the preceding discussion that:

$$|\widehat{A}_{ij} - A_{ij}| \leq \varepsilon \sum_{k=1}^n Q_{kj} \cdot \langle (D^o)_{k,:}, V_{i,:} \rangle + 2\varepsilon n M \tag{111}$$

$$|\widetilde{B}_{ij} - B_{ij}| \leq \varepsilon \langle P_{:,i} + X_{:,j}, 1^n \rangle + \varepsilon^2 n + 2\varepsilon n M_j^{(X)} \tag{112}$$

Thus by the triangle inequality we get the desired error guarantee. $\square$

---

**Algorithm 9** Estimating $D^K$: Putting it all together

---

$\widetilde{B} \leftarrow$ COMPUTE$-B(Q, K, V, D^o, \varepsilon)$
$\widehat{A} \leftarrow$ COMPUTE$-A(Q, K, V, D^o, E, \Sigma, \varepsilon)$
**return** $\widehat{A} - \widetilde{B}$.

---

## G   VECTORIZED IMPLEMENTATION OF THE FORWARD PASS

We present the vectorized implementation of $k$NN Attention that we used in our experiments. This is based on Theorem 2.5.

```
1  import torch
2
3  # This function calculates the attention mechanism in the forward pass.
4  # Inputs:
```

```python
5  # - Q: A tensor of shape (b,h,n,d) containing the query vectors.
6  # - K: A tensor of shape (b,h,n,d) containing the key vectors.
7  # - V: A tensor of shape (b,h,n,d) containing the value vectors.
8  # - k: The number of vectors to attend to via kNN methods.
9  #
10 # Note that b is the batch size, h is the number of heads,
11 # n is the sequence length, and d is the dimension of the vectors.
12 #
13 # Outputs:
14 # - A tensor of shape (b,h,n,d) containing the output vectors.
15 def attn_forward_batched(Q, K, V, k):
16     B,H,N,D = Q.shape
17
18     output = torch.zeros(B,H,N,D, dtype=torch.float32)
19     for b in range(B):
20         for h in range(H):
21             # Get the top k indices of Q[b,h] @ K[b,h]^T and the scores.
22             scores, S = topk(Q[b,h,:,:], K[b,h,:,:], k, masking=True)
23             scores = scores * (1 / math.sqrt(D))
24
25             # Calculate the denominator.
26             M = torch.max(scores, dim=1)[0]
27             exp_scores = torch.exp(scores - M.unsqueeze(1))
28             denom = torch.sum(exp_scores), dim=1).unsqueeze(1)
29
30             # Calculate the numerator.
31             # Vbh[S].shape = (N,k,D)
32             numerator = torch.bmm(exp_scores.unsqueeze(1), V[b,h,S])
33             numerator.squeeze()
34
35             output[b,h] = numerator / denom
36
37     return output
```

Listing 1: kNN Attention - Vectorized

## H    SAMPLES FROM FINE-TUNED GPT2-XL

Below we give some samples from our fine-tuned GPT2-XL using fast approximate $k$NN attention:

- **Prompt: My favorite thing in the world is:** *Just Cause 3. And I couldn't stop playing it. It's a mess. I'm not sure how to describe it. But it's just so fun. When you play it, you know instantly that this is a very different kind of game, one that takes place in an open world of sorts, and you have a bunch of different characters. It's epic, and it's open world, and you can play as a bounty hunter, or you can play as a doctor, or a cowboy, or you can play as a soldier. You have a buddy system, and you can go to another character's story mission if you want. This is a great game, and I'm excited to see what the developers do with it.*

- **Prompt: I like being in nature because** *it's so natural, and because it's always changing. Nature is a friend and a companion, and I'm so thankful every time I'm able to go out into the outdoors. The second thing that I love about being in nature is the comfort and energy that you can feel! I feel so happy and healthy on the trail when I'm interacting with nature. I don't have to think about what I'm wearing, or what I'm doing, and I can just be in nature in my element. I also feel great being out in nature because I'm not spending all day at my desk, working in a office. I get to meet new people along the trail and can spend time talking to them.*

