# OpenReview forum: "kNN Attention Demystified: A Theoretical Exploration for Scalable Transformers"
_ICLR.cc/2025/Conference — ICLR 2025 Poster_

### Official Review · Reviewer_jxN9 · 2024-10-31

**Soundness:** 4
**Presentation:** 4
**Contribution:** 4
**Rating:** 8
**Confidence:** 4

**Summary:**

This paper provides a theoretical analysis of the KNN-nearest-neighbor sparse attention techniques for efficient attention computations. In that setting, every token attends only to its k nearest neighbors. By leveraging developed theoretical framework, the Authors propose a novel algorithm for the sub-quadratic time approximation of the self-attention gradients for efficient Transformer-training (default computations involving attention modules in Transformers require quadratic time in the sequence length, a prohibitively large time complexity for longer input sequences). The conducted analysis leverages an interpretation of the attention mechanism as computing the average value vector with respect to the softmax distribution defined on all the keys. Experimental evaluations confirms Authors' theoretical findings.

**Strengths:**

Great paper, that sheds new light on sparse attention mechanisms leveraging the notion of the kNN-graph. The probabilistic interpretation of the attention mechanism is actually well-known in the literature, yet it is very elegantly applied here to conduct a rigorous theoretical analysis of the method. New algorithm for the sub-quadratic computations of the attention gradients is yet another contribution of very practical impact. The idea to approximate the expectation coming from the probabilistic interpretation of the attention module via lazy Gumbel Noisy sampling is yet another beautiful insight that the paper provides.

**Weaknesses:**

The paper might further benefit from placing newly developed sparse attention mechanisms in the context of other efficient attention methods, e.g. those based on low-rank linear attention. It is also not clear how to choose the optimal value k, since, as the Authors explain, practically optimal value k is often significantly smaller than \sqrt{n}.

**Questions:**

1. The Authors cast it in the paper as an open question, yet I wonder whether they have any intuition how the optimal values of k can be derived in the more mathematically principal way (rather than just empirically), by leveraging the theoretical framework that was already developed in the paper.

2. Did the Authors try to apply proposed in the paper sub-quadratic attention-gradient algorithm for other modalities than text, e.g. in the context of the ViTs ?

---

> ### Author Response · Authors · 2024-11-15
> **Response to Reviewer jxN9**
>
> We thank the reviewer for their time and thoughtful examination of our work! In response to the reviewer’s comments:
>
> 1. **Comparison to other sparse attention mechanisms, like low-rank linear attention:** We agree with the reviewer that placing $k$NN attention in the context of sparse attention mechanisms in general is a very valuable perspective. We believe that it is a very interesting question to perform a thorough comparison between $k$NN attention and other sparse attention methods, both in terms of theoretical guarantees and in the practical sense. Surveys do exist that perform this kind of comparisons, but the field of sparse transformers is continuously growing. In addition, $k$ NN attention has not found its place within this context yet, partly because of the lack of theoretical guarantees for it. As per the reviewer’s suggestion, we have included a small discussion of this topic in our conclusion.
> 2. **The gap between theory and practice for setting** $k$: Explaining this gap between theory and practice is definitely an interesting question. One natural explanation, that may also be a bit unsatisfactory, is that the correct choice of $k$ depends on the data, i.e. the $Q,K,V$ vectors themselves. It is possible that under certain assumptions about the distribution of those vectors one can show theoretically that an asymptotically smaller choice of $k$ also suffices. For instance, our experiments are performed with $Q$ and $K$ containing random vectors whose components are independently chosen from each other, but real datasets do not typically enjoy such independence. Further, the variance of these vectors might be a factor to consider. It is an interesting question to understand whether a specific property of the input vectors makes it easier or harder to sample from the corresponding softmax distribution. This question could also have application in other areas of Deep Learning, such as sampling and outputting a class in a MLP, as Mussmann et. al originally designed the lazy Gumbel sampling method to achieve.
> 3. **Applying to other modalities**: That is an excellent point and one we’d love to try in the future! So far, the $k$ NN attention framework has primarily been applied for text and image generation, but more it is possible that it benefits some modalities more than others!

---

> > ### Comment · Reviewer_jxN9 · 2024-11-25
> > **response**
> >
> > I'd like to thank the Authors for additional comments. In particular, I found additional comments on the correct choice of k very useful. I also appreciate that the Authors have made edits in the manuscript to address some of my comments. Thus I decided to keep my score.

---

### Official Review · Reviewer_AXc3 · 2024-11-03

**Soundness:** 3
**Presentation:** 2
**Contribution:** 2
**Rating:** 5
**Confidence:** 3

**Summary:**

This paper provides a theoretical analysis of $k$-NN attention, which is often explored as a method to reduce the quadratic time complexity of self-attention in Transformers.
Specifically, it demonstrates that by combining $k$-NN with lazy Gumbel sampling, an unbiased estimate of self-attention can be obtained within an $\epsilon$-multiplicative error with sub-quadratic time and space complexity.
Additionally, the paper proposes a method for approximating the gradient of self-attention within $\epsilon$-additive error using random walk simulation, also achieving sub-quadratic time complexity.

**Strengths:**

- The paper excellently ties the use of $k$-NN in self-attention to the context of lazy Gumbel sampling by framing self-attention as an expectation calculation.
- Clear pseudocode is provided for each algorithm.
- The authors have made their experimental code publicly available.

**Weaknesses:**

- **Lack of originality**
  + At the beginning of Section 2.1, the authors claim that the first contribution of this paper is interpreting the softmax computation in self-attention as an expectation calculation. However, this interpretation itself is not a novel idea. For example, see [1].
  + Much of Section 2 simply applies results from [2] to the computation of self-attention. The authors should cite [2] in Theorems 5, 7, and 8, as well as in Lemma 6.
- **Presentation issues**
  + The reference to [2] is listed as an arXiv preprint, but it was accepted at Uncertainty in Artificial Intelligence 2017.
  + In equation (3), the dot is used inconsistently between $q_i^\top \cdot k_k$ and $q_i^\top k_s$.
  + The time complexity in Theorem 2 should be clarified as time complexity in expectation.
  + The citation format on line 97 of page 2 should be consistent with others.
  + "BIN" in Algorithm 1 is not defined.
  + The term "kNN index" lacks sufficient explanation.
  + The sentence "if we assume that..." on line 222 of page 5 was unclear to me; further clarification would be appreciated.
  + It would be helpful to clearly indicate which parts of Algorithm 2 constitute pre-processing.
  + Theorem 8 seems to rely on Theorem 3.5 from [2], which requires the assumption that $V_{sj}$ is bounded. If, like Theorem 2, the assumption $\\|V\\|_{\infty} = O(\log n)$ is used, this should be explicitly stated in the theorem.
  + The reference to Figure 3(a) and (b) is ambiguous; it seems to refer to the figure at the top of page 9. If so, the figure at the bottom of page 9 should be renumbered as Figure 4.
  + Numbered and unnumbered equations are mixed inconsistently.

[1] Kratsios, Universal Regular Conditional Distributions. 2021.
[2] Mussmann et al., Fast Amortized Inference and Learning in Log-linear Models with Randomly Perturbed Nearest Neighbor Search. 2017.

**Questions:**

- In Sections 2.2 and 2.3, two approximation methods are proposed. Can their performance be compared directly?
- In Section 4.1, the experiments are conducted with the matrices $Q,K$, and $V$ sampled from a uniform distribution. How would the performance change if $Q,K$, and $V$ were sampled from more biased distributions, such as those encountered in real-world data?
- In Figure 3(a), why does the computational time for $k=n^{1/4}$  outperform $k=\sqrt{n}$? Also, what does "Brute Force" refer to in this figure?
- In Section 4.2, would it be possible to include a comparison of computation speeds?

---

> ### Author Response · Authors · 2024-11-15
> **Response to Reviewer AXc3**
>
> We thank the reviewer for their thoughtful response and their time. We address their comments below:
>
> 1. **On the originality of our contributions.**
>     1. **The expectation reformulation of attention:** We agree with the reviewer’s comment and  thank the reviewer for pointing us to this part of the literature. We have updated our writing to make it clear that use of such a reformulation is not new to our work. However, our contributions in analyzing $k$ NN attention and providing approximation algorithms for attention gradients nevertheless make novel use of this mathematical tool in the field of sparse attention mechanisms.
>     2. Our contributions can be split into two categories:
>         1. A theoretical framework for $k$NN Attention: We give the first theoretical analysis of the $k$ NN attention paradigm, by utilizing in a novel way the lazy Gumbel Sampling technique of Mussmann et al. and the connection between $k$NN, MIPS and LSH.
>         2. Novel approximation algorithms for attention gradients: we make novel use of techniques from the literature of approximation and randomized algorithms to give sub-quadratic algorithms that approximate the attention gradients.
>     3. *Citing [Mussmann et. al] in Theorems 5,7,8 and Lemma 6*: We have added additional citations. Thank you for pointing that out.
> 2. **Presentation issues**: Thank you for finding these important issues. We have addressed all of them in our updated draft. We point out the following:
>     1. Explaining the sentence fragment: *if we assume that the construction runtime is slightly larger than linear and the query time slightly larger than k*: constructing a kNN data structure takes time slightly super linear, and returns the k nearest neighbors in time proportional to $k$ roughly. We are being loose with this guarantee because we want to remain agnostic with respect to which $k$ NN data structure we choose and what its specific guarantees are. A specific example with using LSH is shown in the Appendix, but other data structures to solve this problem exist as well!
>     2. $||V||_\infty$ should be bounded in Theorem 8
>         - Thank you for catching this! We updated our theorem statement to include this assumption.
> 3. **On the experiments:**
>     1. *Experiments are conducted with respect to uniform distributions? What happens when we use real world data?*
>         - This is a really good question, and the reason why we decided to investigate the quality of approximation when training fine-tuning a real model. Our experiment in Section 4.3 shows that even when the data is not sampled from uniform distributions, the approximation method still obtains a really low average error (although the maximum error per attention entry seems to increase over the iterations)
>     2. *Why does $k=n^{1/4}$ outperform* $k=\sqrt{n}$*?*
>         - In general, the smaller the $k$, the better the performance. However, the concern is that the approximation quality will severly degrade if we set $k$ to be too small. Our theory guarantees that $k=\sqrt{n}$ gives really good approximation guarantees, but in practice we find that setting a lower $k$ also works well. Investigating this gap is a really interesting open question and we believe it is data-dependent!
>         - Brute-Force just refers to the regular quadratic attention algorithm.
>     3. *Speed comparison of backpropagation attention:* our implementation for our gradient approximation algorithms has not been vectorized or parallelized, so the gains will be visible only for really large values of $n$. As per the reviewer’s instructions, we have repeated the experiment for $N=10000$ and observe the following differences:
>         - Time taken for naive grad: 2.164 seconds vs time taken for fast grad: 0.904 seconds
>     4. *Comparing the performance of the up-weighting and median-of-means estimators*:
>         - Thank you for pointing this out! The approximation error is similar between the two, but the performance for median-of-means as implemented is not optimal. That’s because that implementation is harder to vectorize and optimize. We opted to use the up-weighting estimator in our experiments instead.
>
> Please let us know if there are any further questions!

---

> > ### Author Response · Authors · 2024-11-25
> >
> > Hello,
> >
> > Thank you again for your comments. We wanted to check-in one more time before the discussion phase deadline to ensure that your concerns on our presentation and soundness of results have been addressed.
> >
> > Thank you!

---

> > > ### Comment · Reviewer_AXc3 · 2024-11-25
> > >
> > > Thank you for your reply. Below, I have listed my comments on your rebuttal.
> > > 1. The revised paper has made it much clearer what has already been established in prior work and what constitutes the contributions of this paper. Thank you for the revision.
> > > One point of concern is that Lemma 2.1 and Theorem 2.3 should also cite [1].
> > > While Lemma 2.1 provides a new proof, the statements of Lemma 2.1 and Theorem 2.3 were already shown in [1].
> > > To ensure proper recognition of prior work and to avoid potential confusion for readers, it may be helpful to mention near Lemma 2.1 and Theorem 2.3 that "the statements themselves were already shown in [1], but we provide a more concise proof" or something along these lines.
> > > Currently, the sentence just before Lemma 2.1 ("Our simplified proof ... rather than the original exponential-based analysis.") is not clear to first-time readers, and this clarification would help significantly.
> > > 2. I have confirmed that the presentation issues I previously mentioned have been addressed. Thank you for these improvements. Below are a few additional points I noticed:
> > >     - Several reviewers have raised questions about the definition of $T$. One reason for this confusion could be that $T$ is used before its definition is explained. Additionally, the use of $O(T)$ notation might also contribute to the issue. Big O notation is used to describe asymptotic behavior, and it is important to specify which variable is approaching a limit. In this case, I assume you are considering the limit as $n \to \infty$, so it would be clearer to write $T(n)$ or $O(T(n))$ to indicate that $T$ is a function of $n$.
> > >     - This is rather a question, but on the left-hand side of Equation (5), it seems that the multiplication of the variance and the expectation is upper-bounded by $B \cdot O_{ij} \cdot O_{ij}^{-2} = B \cdot O_{ij}^{-1}$. Where did the $O^{-1}_{ij}$ term go?
> > >
> > > Overall, the idea of connecting $k$NN, $k$-MIPS and Gumbel noise sampling is excellent.
> > > However, several pages of the paper are spent restating results from prior work, which makes it less clear what the main contributions of this study are. The bottom-up structure of the paper further adds to this issue (for example, Theorem 2.1, which applies median-of-means boosting, is the first theorem introduced in the paper, even though it is less essential compared to the main contributions of this work, and the authors ultimately state that it is not their preferred implementation for experiments).
> > > Additionally, the experimental results leave some open questions, such as the optimal choice of $k$ for $k$NN Attention, which could be further explored.
> > > For these reasons, I will keep my score as it is.
> > >
> > > **Reference**
> > >
> > > [1] Mussmann et al., Fast Amortized Inference and Learning in Log-linear Models with Randomly Perturbed Nearest Neighbor Search. 2017.

---

> > > > ### Author Response · Authors · 2024-11-25
> > > >
> > > > Thank you for your time! We wanted to briefly continue the discussion, to address some comments.
> > > >
> > > > - We agree that [Mussmann et al., 2017] should be cited in Lemma 2.1 and Theorem 2.3. We will update the manuscript accordingly and emphasize in the main text our contribution to simplifying the proof of Lemma 2.1.
> > > >
> > > > - Indeed, we implicitly assume that $O_{ij}$—the attention output—is lower-bounded by a constant. Thank you for pointing out this omission. This assumption is reasonable since for terms approaching zero, a multiplicative approximation becomes both expensive and unnecessary; in such cases, an additive approximation suffices. We will clarify this in the text promptly.
> > > >
> > > > - We understand that the bottom-up structure may seem unconventional and that reviewing prior work might obscure our main contributions. This presentation was intended to contextualize and unify the algorithmic ideas for clarity. However, we acknowledge your concern and will work to emphasize our main contributions—outlined in Section 1.1—more prominently.
> > > >
> > > > - While median-of-means boosting provides stronger theoretical guarantees, it is less suitable for fast matrix multiplication on hardware, which is why it is not our preferred implementation. We believe the highlight of our work lies in the theoretical framework for $k$-NN attention and our gradient estimation algorithms, rather than the experimental results.
> > > >
> > > > Thank you once again for your valuable feedback, which has significantly improved our work.

---

### Official Review · Reviewer_HavR · 2024-11-09

**Soundness:** 2
**Presentation:** 1
**Contribution:** 3
**Rating:** 5
**Confidence:** 3

**Summary:**

This paper studies the theoretical aspects of k-Nearest-Neighbor (kNN) attention (Roy et al., 2021) and addresses the challenge of quadratic complexity in self-attention.

It reformulates self-attention as expectations over softmax distributions, leveraging Lazy Gumbel Sampling (Mussmann et al., 2017) for efficient approximation. Then, novel sub-quadratic algorithms are introduced for approximating self-attention gradients using Markov Chain methods.

**Strengths:**

- The paper presents a theoretical analysis of kNN attention (Roy et al., 2021), connecting it to Gumbel noise sampling and Markov Chain methods for efficient gradient approximation.

- I think the paper is well-structured.

- Additionally, the author provides empirical experiments that demonstrate scalability and effectiveness in reducing computational and memory overhead.

**Weaknesses:**

- The authors need to clarify their contributions in the introduction and compare the theoretical novelty to Mussmann et al. (2017). The differences between this work and prior works are not clear until one reads the algorithms and theorems in detail.

- Many theorems are presented without proofs or discussions (please see my questions below). This is problematic as even if the proofs are clear to the authors, they should provide proper references or detailed discussions on the theorems.

- The literature contains many randomized attention methods, such as Nyströmformer and Skyformer. These should be discussed in the related works. Adding benchmarks from these methods would also be useful.

- In the experiments section, some parts need clarification (please see my questions). For example, the authors used "kNN" as a legend, but it is unclear whether this refers to standard kNN or their modified version. Similarly, in several places, they mention "our algorithm" without specifying which algorithm they are referring to.

**Questions:**

1) Could you please clarify the theoretical novelty? For example:
- Is Theorem 4 a direct consequence of Median-Of-Means Boosting?
- Does Theorem 8 (main result) rely heavily on Mussmann et al. (2017)?

2) Where is the proof of Theorem 2? Is it a direct consequence of another specific theorem?
3) In Theorem 4, what does $O(T)$ refer to? A discussion on the different parameters in the theorem is needed.
4) In Theorem 28, how large can $\rho$ be? Does it approach 1?
5) Where is the proof of Theorem 29? Is it a direct consequence of (Alman & Song, 2024a; b)?

6) For notation, please clarify Gumbel $(a, b)$ and Bin $(a, b)$.
7) In my opinion, the statement and proof of Theorem 5 are vague. Are you trying to say there exists an index \(\hat{j}\) that gives the maximum, and can you provide a proof for this?
8) In Lines 184–189, what is the Moment Generating Function of the Gumbel distribution used in the proof of Lemma 6?
9) Where is the proof of Theorem 7? What do you mean by "it is easy to derive from Algorithm 1"?
10) In Theorem 7, how strong is the assumption $k = \sqrt{n}$?

11) Algorithm 2 takes $\epsilon$ and $\delta$ as parameters, but I couldn't find them in the algorithm's steps. How are they used? Also, what is the range of these parameters?

12) Theorem 8 is unclear. Why do we need $k$ and $\ell$ to satisfy two inequalities and then set them equal from another inequality? Why not use only the equality case? Considering $\epsilon$ and $\delta$ between 0 and 1, for larger $\ell$, we have:
$
k \geq \sqrt{\frac{8n^2 \varepsilon^{-2} \log(4/\delta)}{\ell}}.
$
However, this assumption seems unrealistic as $k$ scales with $n$.

13) Some parts of Algorithm 3 need clarification. In Line 2, "lg" should be "log". Why should $N$ be selected in that way? This requires more explanation. Also, what is $1^n$?

14) In Line 246, which algorithm are you referring to?
15) It is unclear how the total runtime in Line 224 was obtained. Is this result similar to Theorem 4 under comparable assumptions?

16) What is Figure 3(a)? I couldn’t find it.
17) What is the error definition in the discussion on "Efficiency of kNN Attention" (Lines 421–425)?
18) What do you mean by convex and non-convex cases? In both settings, the attention approximation problem is non-convex due to softmax and the product of $Q$, $K$, and $V$.

19) In the experiments section, you refer to “kNN” and “our algorithms” when comparing them with exact gradients or attention mechanisms. It is still unclear what "kNN" and "our algorithms" refer to in the context of this paper.

20) What specific gains can be observed from the experiments in Sections 4.2 and 4.3? Should we expect time improvements in these sections, for example, in the NanoGPT case? How can we precisely measure the efficiency of the proposed methods, especially for self-attention gradient computation?

---

> ### Author Response · Authors · 2024-11-15
> **Response to Reviewer HavR**
>
> We thank the reviewer for their time and insightful comments! We really appreciate their questions and address them below:
>
> 1. **Our paper’s contributions and clarification of results:**
>     1. A novel theoretical framework for $k$NN Attention, leveraging lazy Gumbel Sampling (Mussmann et al., 2017) and the link among $k$NN, MIPS, and LSH
>     2. New sub-quadratic approximation algorithms for attention gradients, inspired by approximation algorithm techniques
> 2. **Comparing results to previous work:** As per the reviewer's instructions, we added wording to emphasize how prior methods have theoretical guarantees for sparsifying attention, unlike $k$NN attention
>     1. **Novelty on top of the work of Mussmann et al:** While we apply their $k$NN framework for estimating softmax expectations, our work is the first to use it in sparse transformer analysis (Theorems 4 and 8). Additionally, we incorporate diverse analytical tools to develop gradient approximation algorithms.
> 3. **Mathematical rigor and detailed proofs.** We provide full proofs of our Lemmata and Theorems, with the exception of results we borrow from the literature. We have chosen a way of presentation that seeks to derive mathematical results incrementally from first principles, before ultimately arriving at the theorem statement. We believe this to be a style of presentation that promotes readability, but we realize that it might also lead to confusion. Hence, we have heeded the reviewer’s instructions to add more navigational structure between our theorems and proofs. Specifically:
>     1. Theorems 2 and 3 are formally proved in Sections 2 and 3.
>     2. Theorem 29 is proved in Appendix E.
>     3. The proof of Theorem 5 is indeed that short: the Gumbel max trick requires that we pick the index with the largest Gumbel noise. This index will necessarily lie in $S_i \cup T_i$ by construction, since the number of Gumbels greater than $B$ is binomially distributed (see Alg. 1). We hope that clarifies things!
>     4. Theorem 7’s proof is the discussion of Section 2
> 4. **Related literature:** We thank the reviewer for pointing out more relevant sparse attention architectures. We have added the necessary citation to Skyformer.
> 5. We use the term $k$NN attention to refer to Algorithm 2. We have updated it to add clarity.
> 6. **What does $O(T)$ refer to?** It is a placeholder value to be determined on a subsequent theorem (Section 2.2). We have added this clarification.
> 7. **How large can $\rho$ be in Theorem 28? Can it approach** $1$? In typical LSH implementations, $\rho$ is a small constant, typically within the range $(0.25,0.4)$. The concentric LSH constructions amplifies this by a small constant, which we do not resolve as it can be data-dependent.
> 8. **Missing notation:** Gumbel $(a,b)$ is given in Definition 17, and we have added the definition of Bin $(a,b)$.
> 9. **Question 8:** The MGF is given in Lemma 18.
> 10. **Question 10:** The $k=\sqrt{n}$ assumption is crucial to establishing our theorem because it balances the terms $k$ and $n/k$.
> 11. **The role of** $\varepsilon,\delta$: As per the reviewer’s suggestion, we have clarified that all our algorithms are randomized, Monte Carlo algorithms that can fail with probability at most $\delta$.
> 12. **Question 12:** Theorem 8 describes what conditions $k$ and $\ell$ must satisfy to get the additive error guarantee. Because both $k$ and $\ell$ control the final time complexity, we must set them to be equal.
> 13. **Question 13:**
>     1. *Why is $N$ selected this way in line 2?* Equation (22) contains the choice for $N$ so that the Hoeffding bound is satisfied. Hope this helps!
>     2. $1^n$ = vector of $n$ $1$s. This is defined after equation (30).
> 14. **Question 15:** As per the reviewer’s suggestion we have added more clarification of how this runtime is obtained: Instantiating Theorem 4 with $T = O(\sqrt{n})$ due to the choice of $k=\sqrt{n}$ gives $O(nT) \approx O(n^{3/2})$ runtime.
> 15. **Question 17:** Corrected: we refer to *mean, absolute error.*
> 16. **Question 18:** We agree that the approximation problem is inherently non-convex. The focus was on $\phi$'s convexity, which we use in experiments to optimize $\phi(\text{Att}(Q, K, V))$ with gradient descent. Our approximation algorithms provide $\partial \phi / \partial Q$, so convergence depends on $\phi$'s convexity. Convex $\phi$ ensures convergence; for non-convex $\phi$, guarantees are weaker. The reviewer rightly points out that solving this remains complex, and the role of approximation in this setting needs further investigation.
> 17. **Question 20: Efficiency of Experiments**
>     - Previous work shows that $k$NN methods reduce memory use during inference and fine-tuning. We confirm this empirically, noting faster computations and lower memory usage in our framework.
>     - Sparse and approximate attention can degrade quality, but Section 4.3 demonstrates that this effect can be controlled (as measured by perplexity).

---

> > ### Author Response · Authors · 2024-11-25
> >
> > Hello,
> >
> > Thank you again for your comments. We wanted to check-in one more time before the discussion phase deadline to ensure that your concerns on our presentation and clarification of results have been addressed.
> >
> > Thank you!

---

> > > ### Comment · Reviewer_HavR · 2024-11-27
> > >
> > > Dear Authors,
> > >
> > > The renumbering of the theorems and lemmas has made it challenging for me to locate my previous comments. Additionally, I am finding it difficult to carefully track the revisions made to the paper.
> > >
> > > Could you kindly assist me in identifying the specific changes (at least for the theorem, lemma, and corollary numbering) so that I can provide a detailed and careful response?

---

> > > > ### Author Response · Authors · 2024-11-27
> > > >
> > > > Apologies for the confusion due to the re-numbering. Our changes to the manuscript mainly involved small modifications in the ordering of presentation to aid in understanding. We hope that the following is helpful:
> > > > * Theorems 2 and 3 >> Theorems 2.4 and 3.1 (Moved after the intro so that they can be placed alongside their proofs)
> > > > * Theorem 5 >> Thm 2.2 (proof is included in page 4)
> > > > * Theorem 7 >> Thm 2.3 (proof is in preceding discussion)
> > > > * Theorem 4 >> Thm 2.1 (we instantiate it with $T = \sqrt{n}$)
> > > > * Theorem 8 >> Theorem 2.5. (use of existing softmax approximation in the attention calculation. Sub-optimal method in theory, but useful for experiments)
> > > > * Lemma 18 >> Lemma A.2. (gumbel distribution properties)
> > > > * Theorem 29 >> Theorem E.1 (proved in Appendix E)
> > > >
> > > > The remaining points in our response should be independent of the numbering. However, please let us know if there is any additional clarification you need.

---

> > > > ### Author Response · Authors · 2024-12-02
> > > >
> > > > Dear Reviewer,
> > > >
> > > > We wanted to reach out one last time before the end of the discussion phase and see if our earlier comment has provided the necessary help in your response. Please do not hesitate to let us know if you need any additional clarifications.
> > > >
> > > > Thank you!

---

### Official Review · Reviewer_WAZ2 · 2024-11-09

**Soundness:** 3
**Presentation:** 2
**Contribution:** 3
**Rating:** 5
**Confidence:** 3

**Summary:**

The submission presents a theoretical framework for *k*-nearest neighbors (kNN) attention, an alternative to standard self-attention where each token attends to only the *k* "closest" tokens instead of all tokens. Additionally, the authors propose approximation methods for self-attention gradients using kNN search.

More precisely, the submission is organized as follows:

- In part 1, the theoretical results are outlined in brief.
- Part 2 focuses on kNN attention, introduced as an approximation algorithm. In Section 2.1, self-attention is reformulated as an expectation, and Theorem 4 presents the primary approximation results, comparing the outputs of true self-attention and kNN attention, both of dimension $n \times d$.
  - Section 2.2 discusses efficient sampling from the softmax distribution (with each empirical distribution corresponding to a row of the attention matrix) via Gumbel sampling.
  - Section 2.3 introduces an alternative method for computing kNN attention outputs, designed to be more compatible with modern hardware.
- Part 3 describes randomized algorithms to approximate self-attention gradients (with derivations in the appendix). These estimations leverage random walk simulations, and Theorem 10 provides a theoretical runtime analysis.
- Part 4 shows experimental results on kNN attention’s efficiency in terms of space and time costs (Figure (a)) and shows approximation error as a function of k (Figure (b)). Figure 3 compares learning curves for standard gradients obtained using backpropagation with those of the proposed approximation in part 3. Figure 4 evaluates perplexity and approximation error on real-world data using nanoGPT.

**Strengths:**

- The problem studied in the paper is well defined in the abstract.
- Understanding the approximation abilities of sparse attention models, of which kNN attention are a special case, is a significant research question.
- Similarly, proposing alternatives to computationally costly gradient computations in Transformers could have a strong impact in the community.
- The code is provided which enables to reproduce the experimental results.

**Weaknesses:**

I do not believe this paper to be ready to be published at ICLR for the following reasons:

Overall clarity:

The paper is generally quite difficult to follow. Importantly, it is difficult to understand what is new in the paper. A clear statement of the contributions would be helpful.

I am adding specific suggestions / questions for improvement below:

- It should be clearly mentioned that Part 1 sketches the results that are later detailed in Parts 2 and 3.
- Whenever a theorem is stated, a *detailed* proof should be attached to it, even if it is in the appendix. I could not find a proof for each theorem (for instance, theorem 10). Please let me know if I am wrong.
- Lemma 1 is hard to understand as is. What is a multiplicative estimate ? If this is detailed in App A, then the remark in l. 77 should be added before the lemma.
- Line 60 is unclear. For a reader encountering this sentence for the first time, the phrase "how to extend the method to approximate the backward pass" is confusing. Please clarify why this is important and what connection it has to kNN attention.
- Mathematically, the paper lacks rigor. For instance, in line 104, is this an assumption on the differentiability of $\phi$, or is it just notation? Also, where are the precise assumptions on the norms of $Q$, $K$, and $V$ stated? These should appear immediately following the theorem.
- The source of the probability $1 - \delta$ in all the theorems (e.g., line 92) is unclear. Please specify the source of randomness; is it due to sampling over the softmax-induced distribution?
- In line 129, what does $T$ represent?
- In line 120, the term $k_k$ in equation (3) is confusing.
- Similarly, equation (5) is unclear because it takes the expectation with respect to $k$, yet there is an index $k$ in the sum. This could be clarified.

Theoretical Contributions:

- In general, I find the results challenging to interpret. How do these results compare to previous work? What are typical values?
- Presenting attention as an expectation of the value matrix is not new. For instance, see *Attention: Marginal Probability is All You Need* (https://arxiv.org/abs/2304.04556). As written, it seems the submission presents this as a novel contribution, which is misleading.
- Line 181: this is a proof sketch, not a full proof. I could not find the complete proof in the appendix.
- The proof of Theorem 4 is also only a sketch. For instance, see the last two lines.

Experiments:

- There should be experiments specifically validating the theoretical bounds. As is, the experiments are rather qualitative.
- I had to make some manual adjustments to get the provided code running. Including a `setup.py` would be helpful.
- I ran your code for the gradient approximation in Figure 3. On my laptop, the approximated gradient takes approximately 100 times longer to compute than standard gradients. Did you observe similar behavior? In which settings do you expect it to offer speed advantages?
- Similarly, I profiled the code from Appendix G against standard attention on a single GPU, using $B$, $H$, $N$, $D = 256$, $8$, $500$, $32$, and $k = 5$. I observed a runtime of 0.5831 seconds for kNN Attention versus 0.0061 seconds for Traditional Attention. If you do not observe similar results, could you provide code showing that your method leads to a speed advantage?

**Questions:**

Please see the questions in the previous part.

In addition:

- Regarding the conditions in Theorem 8: are these practically achievable? For instance, can $n$, $d$, $T$, and $k$ be expected to yield satisfying approximation errors?

---

> ### Author Response · Authors · 2024-11-15
> **Response to Reviewer WAZ2**
>
> We sincerely thank the reviewer for their careful examination of our work and their valuable feedback. We address the reviewer’s comments and questions below:
>
> 1. **Our paper’s contributions and clarification of results:** Our contributions can be split into two categories:
>     1. A novel theoretical framework for $k$NN Attention, leveraging lazy Gumbel Sampling (Mussmann et al. 2017) and linking $k$NN, MIPS, and LSH.
>     2. New sub-quadratic approximation algorithms for attention gradients, inspired by approximation and randomized algorithm techniques.
>
>     Section 1.1 includes an outline of these contributions, as well as an overview of the technical details. According to the reviewer’s suggestion, we have modified the wording to emphasize our contributions.
>
>     - **Comparing results to previous work:**  We outline how prior methods have theoretical guarantees for sparsifying attention, unlike $k$NN attention.
>     - **The expectation reformulation of attention:** We agree with the reviewer’s comment and  thank the reviewer for pointing us to this part of the literature. We have updated our writing to make it clear that use of such a reformulation is not new to our work. However, our contributions nevertheless make novel use of this mathematical tool in the field of sparse attention mechanisms.
> 2. **Mathematical rigor and detailed proofs.** We provide full proofs of our Lemmata and Theorems, with the exception of results we borrow from the literature. We have chosen a way of presentation that seeks to derive mathematical results incrementally from first principles, before ultimately arriving at the theorem statement. That is the case, for example, with Theorems 7, 10 and 29. We believe this to be a style of presentation that promotes readability, but we realize that it might also lead to confusion. Hence, we have heeded the reviewer’s instructions to add more navigational structure between our theorems and proofs.
>     1. The proof of Theorem 5, which the reviewer alluded to, is indeed that short: the gumbel max trick requires that we pick the index with the largest Gumbel noise. This index will necessarily lie in $S_i \cup T_i$ by construction, since the number of Gumbels greater than $B$ is binomially distributed (see Alg. 1). We hope that clarifies things!
> 3. **Multiplicative Estimates in Lemma 1:** This boosting lemma is presented in more detail in the Preliminaries section of the appendix, where the term “*multiplicative estimate”* is defined. According to the reviewer’s suggestion, the initial reference to the Appendix was adjusted for clarity
> 4. **Line 60:** Thank you for pointing out the confusing part about the writing. We have updated to: “whether these methods can yield efficient algorithms for approximating the backward pass.”
> 5. **The role of** $\delta$: As per the reviewer’s suggestion, we have clarified that all our algorithms are randomized, Monte Carlo algorithms that can fail with probability at most $\delta$.
> 6. **What does** $T$ **represent?** It is an upper bound to the time it takes to sample from $D_i$. We have added an additional explanation for $T$ in the following paragraph.
> 7. **Avoiding** $k_k$**:** thank you for pointing this out. We have clarified the notation.
> 8. **On the purpose of the experiments:** Since our results are primarily theoretical, we found it very difficult to be precise in evaluating the theoretical bounds. The use of $O$-notation hides constants that will appear in the experiments and can degrade the quality of our conclusions. Our experiments study both the validity of our theory and the behavior of our algorithms in practice, resulting in numerous new research directions
> 9. **setup.py** Thank you for pointing this difficulty out and for examining our code! We have added a setup.py file to the repo.
> 10. **Efficiency of the gradient approximation:** Our implementation for gradient approximation has not been optimized or vectorized, and that experiment mainly aims to show that the error accumulation in gradient descent does not immediately lead to divergence. However, after repeating the experiment ``experiment_grad_descent.py`` again we find that the approximate gradient runs in time comparable to the naive one, even with small values of $n$. We anticipate the gain in efficiency to be most evident for larger values of $n$:
>     - n=10000: Iteration 2: Fast attention gradients: 0.92 seconds, Naive Attention Gradients: 2.02 seconds
> 11. **Profiling of** $k$**NN attention:** Referencing ``forward_pass_test_batched.py``, it is important to note that the performance gains appear for large values of $n$ (see Figure 3). Also, we expect the performance to look different depending our the setup of the machine the experiment runs in. After running the experiment again in a MacBook Air CPU, with 8GBs of RAM we with n=13000, we observe:
>     - Naive time =  59.50395321846008, Attn time =  8.40250277519226

---

> > ### Comment · Reviewer_WAZ2 · 2024-11-22
> >
> > Thank you for your response.
> >
> > > Mathematical rigor and detailed proofs. We provide full proofs of our Lemmata and Theorems, with the exception of results we borrow from the literature. We have chosen a way of presentation that seeks to derive mathematical results incrementally from first principles, before ultimately arriving at the theorem statement.
> >
> > Can you then please specify which proof is related to which result in the paper ? Which ones are borrowed from the literature ? Thank you.
> >
> > > After running the experiment again in a MacBook Air CPU, with 8GBs of RAM we with n=13000, we observe: Naive time = 59.50395321846008, Attn time = 8.40250277519226
> >
> > Could you provide me with the code to reproduce this experiment on my laptop ? Thanks
> >
> > I will consider increasing my score if the authors could provide me with those informations.

---

> > > ### Author Response · Authors · 2024-11-22
> > >
> > > Thank you for your response!
> > >
> > > > Can you then please specify which proof is related to which result in the paper? Which ones are borrowed from the literature? Thank you.
> > >
> > > Theorems 5,7 and 26 are proven in [Mussmann et al, 2017] and we adapt them in the context of computing self-attention in our theoretical framework. Theorems 2,3,4,10,28,29 and 32 are directly related to our results and we give proofs for them as follows:
> > > * Theorem 2 follows directly after Theorems 4, 7 and the discussion on paragraph 2.2.1
> > > * Theorem 3 is just the combined statement of Theorems 10, 29 and 32.
> > > * Theorem 4's proof is given in page 3.
> > > * Theorem 10's proof is given in paragraph 3.1 (the full error guarantee follows after paragraph 3.1.1 and equation (36). The runtime guarantee is proven in paragraph 3.1.2).
> > > * Theorems 28, 29 and 32 are proven in the paragraphs of Appendices C, E and F respectively.
> > > * Lemma 9 is proven in Appendix D.
> > >
> > > We recognize that this presentation style is a bit unconventional, so we will soon post an update of our draft containing precise connections between theorems and proofs. We hope this helps!
> > >
> > > > Could you provide me with the code to reproduce this experiment on my laptop ? Thanks
> > >
> > > Absolutely! We are running the following command line: ```python -m backward_pass.experiment_grad_descent```. We have updated the file in the code's repo to include some timers in the way we used them to measure performance. Using N = 20000, num_iterations = 10, attention gradients take around 10 seconds to compute, whereas the gradients with respect to V take around 0.8 seconds.

---

> > > > ### Comment · Reviewer_WAZ2 · 2024-11-24
> > > >
> > > > Thank you.
> > > >
> > > > I saw you updated the pdf, adding more clarity in the paper, in particular by connecting the mathematical statements to their proofs.
> > > >
> > > > I ran your code on my laptop, I get a 2.5x improvement for your gradient implementation rather than 10x.
> > > >
> > > > > Profiling of $k$NN attention: Referencing forward_pass_test_batched.py, it is important to note that the performance gains appear for large values of $n$ (see Figure 3). Also, we expect the performance to look different depending our the setup of the machine the experiment runs in. After running the experiment again in a MacBook Air CPU, with 8GBs of RAM we with n=13000, we observe:
> > > >
> > > > How can I reproduce this particular experiment ? Thank you.

---

> > > > > ### Author Response · Authors · 2024-11-24
> > > > >
> > > > > Thank you for your response and for taking the time to run our experiment again!
> > > > >
> > > > > It is possible that the exact speed up depends on the machine configuration and resources. Note also that because we have not vectorized and optimized our gradient approximation code, we have not included in our paper experimental results studying the performance boost as compared to the naive gradient computation. We made this choice because the merit of using gradient approximations in training deep neural models is unclear and warrants its own investigation. Rather, we focus on studying the approximation quality of the algorithms in Section 4.2., for which the results can be seen in Figure 3.
> > > > >
> > > > > Finally, we replicated the ```backward_pass.experiment_grad_descent``` experiment again with N = 20000 for 10 iterations. The output we observe on our machine is as follows:
> > > > > ```Iteration: 1
> > > > > Time taken for attention gradients: 20.156655073165894
> > > > > Time taken for fast grad V: 0.8495640754699707
> > > > > Iteration: 2
> > > > > Time taken for attention gradients: 10.584980964660645
> > > > > Time taken for fast grad V: 0.9910299777984619
> > > > > Iteration: 3
> > > > > Time taken for attention gradients: 11.594069957733154
> > > > > Time taken for fast grad V: 0.9713640213012695
> > > > > Iteration: 4
> > > > > Time taken for attention gradients: 14.452584981918335
> > > > > Time taken for fast grad V: 1.050285816192627
> > > > > ...
> > > > > ```
> > > > >
> > > > > Thank you again for your engagement! Please let us know if there are any further questions.

---

> > > > > > ### Comment · Reviewer_WAZ2 · 2024-11-25
> > > > > >
> > > > > > Thank you. I was referring to the attention matrices computations where you report:
> > > > > >
> > > > > > > Naive time = 59.50395321846008, Attn time = 8.40250277519226
> > > > > >
> > > > > > Thanks

---

> > > > > > > ### Author Response · Authors · 2024-11-25
> > > > > > >
> > > > > > > Apologies for the confusion! We can give more details on the forward-pass experiment as well. That experiment is ```python -m forward_pass.forward_pass_test_batched.py```. We ran it again with $n=13000$ and observed similar outputs:
> > > > > > > ```
> > > > > > > B =  10 k =  3
> > > > > > > Naive time =  65.04901885986328
> > > > > > > Attn time =  8.869472980499268
> > > > > > > Naive time =  59.6989369392395
> > > > > > > Attn time =  8.889343023300171
> > > > > > > Naive time =  65.76588797569275
> > > > > > > Attn time =  8.690060138702393
> > > > > > > Naive time =  60.717044830322266
> > > > > > > Attn time =  8.869831800460815
> > > > > > > Mean error =  0.017475314 with std dev =  0.0
> > > > > > > Max error =  6.0020223 with std dev =  0.0
> > > > > > > B =  10 k =  6
> > > > > > > Naive time =  80.37394785881042
> > > > > > > Attn time =  11.750183343887329
> > > > > > > Naive time =  58.98138499259949
> > > > > > > Attn time =  10.925697088241577
> > > > > > > Naive time =  67.75123572349548
> > > > > > > Attn time =  11.87036919593811
> > > > > > > ...
> > > > > > > ```
> > > > > > >
> > > > > > > Note that the 10x speed-up refers to using $k=3$, and the experiment uses multiple values for $k$. With larger values of $k$ the performance boost naturally decreases. We give an analysis of that behavior in Figure 2. That being said, it is also possible again that the precise speed-up is dependent on the machine configuration and resources one uses. Hope that helps!

---

> > > > > > > > ### Comment · Reviewer_WAZ2 · 2024-11-28
> > > > > > > >
> > > > > > > > I am trying to run the code but got an error:
> > > > > > > >
> > > > > > > > "forward_pass/forward_pass_test_batched.py", line 40, in <module>
> > > > > > > >     raise ValueError("kk_labels and kk must have the same length.")
> > > > > > > > ValueError: kk_labels and kk must have the same length."
> > > > > > > >
> > > > > > > > I made sure to git pull before running your command
> > > > > > > >
> > > > > > > > Thanks

---

> > > > > > > > > ### Author Response · Authors · 2024-11-28
> > > > > > > > >
> > > > > > > > > Apologies - we forgot to update the value of $n$: Could you try $n=13000$ for that experiment? That should work! We'll update the code shortly.
> > > > > > > > >
> > > > > > > > > Thank you!

---

> > > > > > > > > > ### Comment · Reviewer_WAZ2 · 2024-12-01
> > > > > > > > > >
> > > > > > > > > > Dear authors,
> > > > > > > > > >
> > > > > > > > > > Thank you for your responses. I am raising my score to 5.

---

### Official Review · Reviewer_SjhM · 2024-11-12

**Soundness:** 3
**Presentation:** 3
**Contribution:** 3
**Rating:** 8
**Confidence:** 4

**Summary:**

This paper theoretically analyzes kNN attention, which improves on the quadratic complexity (with respect to context length, $n$) of traditional full attention. The authors do this by viewing the self-attention matrix as an expectation over multiple softmax distributions, and then use the Gumbel-Max trick, along with the concentration properties of the Gumbel distribution, to efficiently approximate the original self-attention. This lazy Gumbel sampling, combined with k-MIPS, results in a total runtime of $O(n^{1.5})$. Additionally, the work approximates the attention gradients using length-1 random walk sequences, reducing the naive complexity from $O(n^2d)$ to $O(nd^2)$, providing high-probability approximation bounds.

**Strengths:**

The paper focuses on the important and relevant topic of the quadratic complexity of attention, given the ever-increasing context length in LLMs, and provides a rigorous theoretical analysis behind the empirical performance of kNN attention.
The paper is generally well-written, easy to read, and the ideas are clearly organized and discussed throughout. I enjoyed reading it, and liked how the authors first decompose the attention matrix as expectations over softmax, then use median-of-means boosting to achieve high-probability approximation bounds and runtime complexity bounds. The use of the Gumbel-Max trick and the concentration properties of the Gumbel distribution is also cute, ultimately leading to the gain over quadratic attention.

The use of 1-step random walks over the transition matrix P to approximate matrix-vector products (giving attention gradients in this case), although known, is also pretty nice. Overall, I appreciate how different well-known ideas are effectively combined.

**Weaknesses:**

I don’t have many weaknesses to point out, but I do have some questions/points I’d like to raise/discuss:


—The experiments in Figure 3b) show that for $k>n^{⅛}$, kNN attention performs quite well, and there doesn’t seem to be much need for $n^{1/2}$ (as suggested in your theory). I understand you’ve mentioned this as potential future work in the conclusion, but why do you think this is the case? As far as I understand, the choice of $k=\sqrt{n}$ in your theory arises because you want to balance the samples outside $S_i$​, which could, in expectation, ruin the top score after the addition of Gumbel noise, and the accuracy of lazy Gumbel sampling. This factor of $n/k$ also appears in the discussion in (Routing Transformers, Roy et al.), where the complexity is $O(nkd+n^2d/k)$, and $\sqrt{n}$ is the optimal choice. What do you think explains this significant gap?


—(Related to the above) For kNN attention without median-of-means (Sec 2.3), you randomly sample outside the top $k$ similarity products and upweigh them to capture the tail of interactions, and this is the algorithm used in the experiments. Median-of-means doesn’t consider the tail at all. Do you think capturing the tail behavior is critical to $k≪\sqrt{n}$​ performing well?

—Regarding the experiments in Section 4.2: The true benefit in the backward pass should only show up with large $n$. I understand that training larger models is difficult, but it would be interesting to see what happens with a moderate $n∼1000$ when training with cross-entropy.

—What is $n$ for the final set of experiments (perplexity and approximation error)? Also, for the mean error of the kNN ($k$ vs. $n$) experiment, what is the range of $n$? I couldn’t find these details in the appendix.


—Minor point: There is no figure number for the figure at the top of page 9, which I believe should be Figure 3. Please fix this and the rest.

**Questions:**

Please see weaknesses section.

---

> ### Author Response · Authors · 2024-11-15
> **Response to Reviewer SjhM**
>
> We thank the reviewer for their time and attentive examination of our work. We provide a detailed response to each question and comment raised:
>
> 1. **The gap between theory and practice for setting** $k$: Explaining this gap between theory and practice is definitely an interesting question. One natural explanation, that may also be a bit unsatisfactory, is that the correct choice of $k$ depends on the data, i.e. the $Q,K,V$ vectors themselves. It is possible that under certain assumptions about the distribution of those vectors one can show theoretically that an asymptotically smaller choice of $k$ also suffices. For instance, our experiments are performed with $Q$ and $K$ containing random vectors whose components are independently chosen from each other, but real datasets do not typically enjoy such independence. Further, the variance of these vectors might be a factor to consider. It is an interesting question to understand whether a specific property of the input vectors makes it easier or harder to sample from the corresponding softmax distribution. This question could also have application in other areas of Deep Learning, such as sampling and outputting a class in a MLP, as Mussmann et. al originally designed the lazy Gumbel sampling method to achieve.
> 2. **About the differences between median-of-means and the up-weighting estimator:**
>     1. One major difference between these two estimators is that the up-weighting estimator provides guarantees for additive error, whereas the median-of-means gives a multiplicative error guarantee. It is possible that the analysis changes slightly in ways that make the sample sizes more comparable if we require both errors to be of the same type.
>     2. *About sampling from the tail:* Thank you for the valuable intuition on the nature of the sampling process! The median-of-means estimator does consider the tail of interactions, although in a more repeated fashion: the basic estimator from Algorithm 1 picks $m$ points from the tail, and the median-of-means boosting repeatedly applies that sampling process. On the other hand, the up-weighting estimator only looks at the tail once, which results in a different analysis and theoretical guarantees.
> 3. **On the experiments of the approximate backward pass.** This is a question that we feel is very interesting as well! Our experiments in Section 4.2 serve the purpose of showcasing that our approximation algorithms can indeed be used to optimize a function with tolerable error, but whether approximate gradient estimation is good enough to train a model with reasonable accuracy is an open question. As the reviewer correctly points out, the benefit would only show up with larger values of $n$, where the space and time superiority of approximation is evident against naive calculations. Indeed, that is something we observe for $n \geq 10000$ in our experiments, where we see that approximate gradients outperform the naive computation, both in time and space, by a factor of at least 2. We did experiment with approximate gradients in the *fine-tuning* of a model, but we deemed our results inconclusive, mainly because of the volatility of the outputs. It is a strong possibility that under the correct fine-tuning conditions these algorithms can successfully be deployed in accelerating fine-tuning and even pre-training, but we chose to leave this task to future investigations.
> 4. **What is** $n$ **in the final set of experiments?** This value is equal to $n= 1024$, with $d=768$ as the embedding dimension. We have fixed this in the paper.
> 5. **Figure 3 is not labeled.** This has also been fixed.

---

### Author Response · Authors · 2024-11-22

We thank the reviewers for all their helpful comments and insightful questions. We have addressed these questions in detail in individual responses below.

As noted by more than one reviewer, our presentation style can at times lead to confusion around the relationship between our discussion, our theorems and our proofs. To address these concerns, we have updated our draft to ensure additional clarity in navigating our mathematical statements and their proofs. Now, every stated theorem is precisely accompanied by its full proof, or a pointer to where the proof can be found in the literature. This is only a small structural change: we have not inserted any new mathematical insight other than the corrections pointed out by the reviewers.

We hope that this clarifies our work more, and we again thank the reviewers for their time.

---

### Meta-Review · Area_Chair_DyiP · 2024-12-22

**Metareview:**

The paper addresses an important and relevant topic of quadratic complexity of attention in Transformers, and provides a theoretical analysis behind the empirical performance of KNN attention. The paper is well-written and the ideas are clearly organized. The authors effectively combine several well-known ideas, such as decomposing the attention matrix as expectations over softmax, using median-of-means boosting to achieve high-probability approximation bounds and runtime complexity bounds, and employing the Gumbel-Max trick and the concentration properties of the Gumbel distribution. The paper proposes new sub-quadratic approximation algorithms for attention gradients, inspired by approximation and randomized algorithm techniques.

**Additional Comments On Reviewer Discussion:**

- The authors emphasized that their main contributions were providing a theoretical framework for kNN attention and proposing novel approximation algorithms for attention gradients.
- They added full proofs of their lemmas and theorems, and made small modifications to the ordering of presentation to aid in understanding.
- They explained that the correct choice of k depends on the data, and that under certain assumptions about the distribution of the data, one can show theoretically that an asymptotically smaller choice of k also suffices.
- They included citations to Skyformer and other relevant sparse attention architectures.

---

### Decision · Program_Chairs · 2025-01-22

Accept (Poster)